# Polar boundary layer bromine explosion and ozone depletion events in the chemistry-climate model EMAC v2.52: Implementation and evaluation of AirSnow algorithm

Stefanie Falk[1,a] and Björn-Martin Sinnhuber[1]

[1]Institute of Meteorology and Climate Research, Karlsruhe Institute of Technology, Karlsruhe, Germany
[a]now at: Department of Geosciences, University of Oslo, Oslo, Norway

**Correspondence:** Stefanie Falk (stefanie.falk@geo.uio.no)

**Abstract.** Ozone depletion events (ODE) in the polar boundary layer have been observed frequently during spring-time. They are related to events of boundary layer enhancement of bromine. Consequently, increased amounts of boundary layer volume mixing ratio (VMR) and vertical column densities (VCD) of $BrO$ have been observed by in situ observation, ground-based as well as air-borne remote sensing, and from satellites. These so called bromine explosion (BE) events have been discussed serving as source of tropospheric $BrO$ at high latitudes, which has been underestimated in global models so far. We have implemented a treatment of bromine release and recycling on sea ice and snow covered surfaces in the global chemistry-climate model EMAC (ECHAM/MESSy Atmospheric Chemistry) based on the scheme of Toyota et al. (2011). In this scheme, dry deposition fluxes of HBr, HOBr, and $BrNO_3$ over ice and snow covered surfaces are recycled into $Br_2$ fluxes. In addition, dry deposition of $O_3$, dependent on temperature and sunlight, triggers a $Br_2$ release from surfaces associated with first-year sea ice. Many aspects of observed bromine enhancements and associated episodes of near-complete depletion of boundary layer ozone, both in the Arctic and in the Antarctic, are reproduced by this relatively simple approach. We present first results from our global model studies extending over a full annual cycle, including comparisons with GOME satellite $BrO$ VCD and surface ozone observations.

## 1  Introduction

Events of near-complete depletion of polar boundary layer ozone are observed frequently during spring-time over both hemispheres (Oltmans, 1981; Barrie et al., 1988; Bottenheim et al., 1986, 2002, 2009). Individual events typically last between several hours to a few days (Strong et al., 2002). The boundary layer ozone depletion events (ODE) are almost certainly related to events of strongly enhanced bromine, so called bromine explosion (BE) events. Enhanced bromine monoxide (BrO) column densities and mixing ratios are regularly observed by in situ observation, ground-based remote sensing (e.g., Network for the Detection of Atmospheric Composition Change, NDACC), and from satellites (e.g., Wagner and Platt, 1998). Data from satellites and additional aircraft campaigns, e.g., ARCTAS, POLARCAT (Choi et al., 2012; Jacob et al., 2010), provide the spatial extend of such events. ODE predominantly occur over the marginal sea ice zone, but sometimes also over inland ice and snow covered regions (e.g., Richter et al., 1998). In addition to their impact on boundary layer ozone, BE events may play an impor-

tant role in mercury deposition and corresponding environmental impacts (Lindberg et al., 2002; Stephens et al., 2012; Toyota et al., 2014b, a). Proposed mechanisms for BE events involve frost flowers on thin sea ice (Kaleschke et al., 2004) and blowing of saline snow on sea ice (Yang et al., 2010). Additional observational evidence for a significant contribution of high wind speeds to BE events has been found over both hemispheres in satellite data (Jones et al., 2009, 2010). Carbonate precipitation in brine at low temperatures has been suggested as efficient release trigger of sea salt bromine to the atmosphere (Sander et al., 2006). However, measurements of $Br_2$ release in dependence of illumination and ozone volume mixing ratio (VMR) from various types of snow and ice indicate that neither sea ice itself nor brine icicles are a major source for $Br_2$ (Pratt et al., 2013). Pratt et al. (2013) have also found that not only snow on sea ice has to be taken into consideration as source of bromine but also snow on land surfaces may contribute. In addition to these natural sources, anthropogenic $NO_x$ emission enhance reactive bromine species in the polar boundary layer (Custard et al., 2015). Recent reviews on the subject are provided by Simpson et al. (2007), Saiz-Lopez and von Glasow (2012), and Abbatt et al. (2012). There has been considerable progress in describing the mechanisms involved in bromine release and boundary layer ODE based on field measurements, laboratory experiments, and process modeling (Toyota et al., 2014b). Regarding the underlying heterogeneous chemical reactions, some similarities can be drawn between the very cold polar boundary layer and the polar upper troposphere - lower stratosphere (UTLS), where polar stratospheric clouds (PSCs) play a major role in halogen activation. In these cold regimes, icy surfaces allow or accelerate reactions, which are impossible or rather slow in gas-phase chemistry. For sustaining catalytic ozone depletion, the activation of halogens through heterogeneous reactions is very important. According to Abbatt et al. (2012), the existing modeling approaches can be grouped into four categories:

- Frost flowers ($\rightarrow$ sea salt aerosol formation),

- bulk ice and snow ($\rightarrow$ $Br_2$ release),

- blowing of saline snow ($\rightarrow$ uplifting of sea salt and aerosol formation), and

- snowpack (photo)chemistry ($\rightarrow$ $Br_2$ release).

Frost flowers covered in high saline brine, are sturdy while fragile in appearance and contribute less to saline aerosol formation and BE events than originally anticipated (Domine et al., 2005). $Br^-$ enriched brine is formed on sea ice through drainage and precipitation of hydrohalite ($NaCl \cdot 2H_2O$) at temperatures below $251\,K$ (Abbatt et al., 2012, and references therein). In the course of summer, most salt is washed out from sea ice. Therefore, multi-year sea ice contains much less salt than first year ice and may be discarded as a major source of BE events. The importance of acidity for reaction kinetics on icy surfaces strongly depends on the involved species. While HOBr uptake on frozen NaBr/NaCl solutions is not dependent on acidity (Adams et al., 2002), uptake reactions of gas-phase $O_3$ are fastest on acidic media (Oldridge and Abbatt, 2011). $Br_2$ as a precursor of BrO is formed in complex heterogeneous photochemistry, which is taking place in the quasi-liquid layer on ice grains in the snowpack (Thomas et al., 2011; Pratt et al., 2013). The rate at which $Br_2$ is released is mainly limited by mass transfer from the atmosphere to snow or ice due to the rapid reaction of HOBr to $Br_2$ (Huff and Abbatt, 2000). Ozone itself has the capacity of triggering auto-catalytic reactions by oxidizing bromine in snow and ice photochemically as well as non-photochemically.

Subsequently, $Br_2$ is released.

On the basis of empirical and modeling results, Toyota et al. (2011) presented a parameterization of $Br_2$ release from bulk ice and snow within the Global Environmental Multiscale model with Air Quality processes (GEM-AQ). GEM-AQ is based on Canada's operational weather prediction model developed by the Meteorological Services of Canada (MSC) for the interaction of atmospheric chemistry with sea ice and snow surfaces. Toyota et al. (2011) have shown that many aspects of observed bromine enhancements and boundary layer ODE in Arctic spring can be reproduced with their simple approach of recycling $HOBr$, $BrNO_3$, and $HBr$ into $Br_2$.

Here we present an implementation of a mechanism based on the work of Toyota et al. (2011) into the ECHAM/MESSy Atmospheric Chemistry (EMAC) model (Jöckel et al., 2010). The mechanism and its integration into the existing submodel ONEMIS (Kerkweg et al., 2006a) are described in Sect. 2. In Sect. 3, results from several one year long integrations of the model with and without bromine release are presented and compared to observations of $BrO$ vertical column density (VCD) from the Global Ozone Monitoring Experiment (GOME) satellite instrument on board ERS-2 (Richter et al., 1998, 2002) (Sect. 3.1) as well as surface ozone observations (Sect. 3.2). We show that many aspects of observations regarding $BrO$ enhancements and ODE are reproduced by this mechanism without any further tuning of parameters. Unlike Toyota et al. (2011), we do not focus on Arctic spring time only but investigate the applicability of the mechanism on a full annual cycle and in both hemispheres.

## 2    Model and experiments

The EMAC model is a numerical chemistry-climate model, based on the 5th generation European Centre Hamburg general circulation model (ECHAM5) (Roeckner et al., 2006) as dynamical core. Various submodels describe atmospheric and Earth system processes and are coupled via the Modular Earth Submodel System (MESSy) (Jöckel et al., 2005). MESSy provides an infrastructure with generalized interfaces for control and coupling of components. Further information about MESSy and EMAC is available from the MESSy project homepage (www.messy-interface.org). MESSy enables for a flexible handling of emissions in EMAC, e.g., prescribed fluxes, concentrations of tracers at the boundary layer or any other given level, or emissions interactively dependent on dynamical atmospheric fields. Latter are treated as online emissions using the submodel ONEMIS (Kerkweg et al., 2006a). ONEMIS provides facility functions for flux to tracer concentration conversions. According to the MESSy philosophy, ONEMIS is separated into a submodel interface layer (SMIL) for unified data handling among different submodels and an implementation layer of the actual emission mechanisms (submodel core layer, SMCL). A recap of the mechanism proposed by Toyota et al. (2011) (Sect. 2.1) and details about its integration into the EMAC model (Sect. 2.2) are given in the following. In Sect. 2.3, scope and setup of a set of test experiments are summarized.

### 2.1    Description of the mechanism

Toyota et al. (2011) assume that at least part of the observed $Br_2$ flux originates from heterogeneous reactions on snow grains in the surface layer of a snowpack (Pratt et al., 2013). These snow grains are considered coated by a $Br^-$ enriched film of liquid water and show a distinct acidity. In this quasi-liquid phase, heterogeneous reactions of $HOBr$ and $BrNO_3$ with either

$Br^-$ and $Cl^-$ can take place:

$$HOBr + Br^- \xrightarrow{H^+} Br_2 + H_2O, \tag{R1}$$

$$BrNO_3 + Br^- \rightarrow Br_2 + NO_3^-, \tag{R2}$$

$$HOBr + Cl^- \xrightarrow{H^+} BrCl + H_2O, \tag{R3}$$

$$BrNO_3 + Cl^- \rightarrow BrCl + NO_3^-. \tag{R4}$$

Interhalogene reactions may convert $BrCl$ into $Br_2$:

$$BrCl + Br^- \leftrightarrow Br_2 + Cl^- \tag{R5}$$

BrCl is partly released to the atmosphere before undergoing this last reaction. In addition, various photochemical gas-, aqueous-, and heterogeneous-phase reactions are taking place in the top layer of the snowpack (for details see, e.g., Pratt et al., 2013, Fig. 2). A list of heterogeneous reactions involving bromine included in MECCA is provided as Supplement S.1. Another reaction pathway oxidizing bromine is triggered by ozone dry deposition. Three surface types, first-year sea ice (FY),
multi-year sea ice (MY), and snow on land (LS), are differentiated. In any case, the respective surface temperature has to be below a temperature threshold $T_{\mathrm{crit}}$. The critical conversion of a dry deposition flux of ozone ($\Phi_{O_3}$) into an emission flux of $Br_2$ (or BrCl) is moderated by an ad hoc molar yield $\Phi_1$, dependent on surface type and illumination. Toyota et al. (2011) have parameterized the above heterogeneous reaction pathways (R1–R4), which transform any of the dry deposition fluxes of $HOBr$, $BrNO_3$, or $O_3$ to $Br_2$, in a simple way taking state-of-the-art knowledge into account:

$$\Phi_1 = \begin{cases} 0.001 & \text{if } dark \text{ FY}, \\ 0.075 & \text{if } sunlit \text{ FY}, \\ 0 & \text{if MY or LS}. \end{cases} \tag{1}$$

I.e., on FY sea ice, only 0.1% of the dry deposition of $O_3$ will be converted into $Br_2$ in case the surface is not sunlit (sun's zenith angle above $\theta_{\mathrm{crit}} = 85\,^\circ$), otherwise 7.5% is converted. No release of $Br_2$ from MY sea ice or LS is assumed. The specific value of $\Phi_1$ has been obtained as best choice by cross-validating modeling results with observed spring-time boundary layer ozone data at Alert, Barrow, and Zeppelin (Toyota et al., 2011, Section 3.1).

The conversion of dry deposition fluxes of $HOBr$ ($\Phi_{HOBr}$), $BrNO_3$ ($\Phi_{BrNO_3}$), and $HBr$ ($\Phi_{HBr}$) is considered independent of illumination. In case of FY sea ice, the snowpack on top is regarded as an infinite pool of $Br^-$ and $Cl^-$. The sum of $HOBr$ and

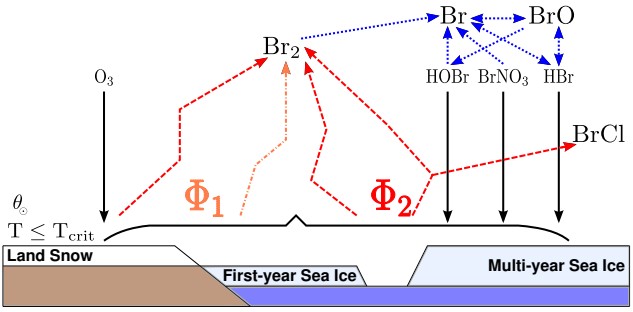

**Figure 1.** Schematic scenario of bromine release from first-year sea ice, multi-year sea ice, and land snow adapted from Toyota et al. (2011) for a temperature threshold $T_{\mathrm{crit}}$. Black arrows denote dry deposition of HOBr, BrNO$_3$, HBr, and O$_3$. Blue doted arrows indicate gas-phase photochemistry. Dry deposition fluxes are recycled into Br$_2$ with respect to a molar yield $\Phi_1$ in case of O$_3$ (dashed orange) and $\Phi_2$ in case of the brominated species (dashed red).

BrNO$_3$ dry deposition fluxes ($\Phi_{\mathrm{HOBr}} + \Phi_{\mathrm{BrNO_3}}$) is fully recycled into Br$_2$. In case of MY sea ice, only the Cl$^-$ pool remains infinite, for Cl$^-$ is about 2 to 3 orders of magnitude more abundant in snow than Br$^-$ (Toyota et al., 2011). The release of Br$_2$ depends on $\Phi_{\mathrm{HOBr}} + \Phi_{\mathrm{BrNO_3}}$ in comparison to the dry deposition flux of HBr. If $\Phi_{\mathrm{HOBr}} + \Phi_{\mathrm{BrNO_3}}$ was less than $\Phi_{\mathrm{HBr}}$ a full conversion of $\Phi_{\mathrm{HOBr}} + \Phi_{\mathrm{BrNO_3}}$ to Br$_2$ is assumed. Otherwise, only half of the difference $\Phi_{\mathrm{HOBr}} + \Phi_{\mathrm{BrNO_3}} - \Phi_{\mathrm{HBr}}$ is recycled

to Br$_2$, the other half is converted to BrCl. For LS, neither Br$^-$ nor Cl$^-$ is available unlimited. Hence, only the smaller of $\Phi_{\mathrm{HOBr}} + \Phi_{\mathrm{BrNO_3}}$ and $\Phi_{\mathrm{HBr}}$ is converted to Br$_2$. The resulting *yield* is summarized in $\Phi_2$:

$$\Phi_2 = \begin{cases} 1 & \text{if FY,} \\ 0.5 - 1 & \text{if MY,} \\ 0 - 1 & \text{if LS.} \end{cases} \tag{2}$$

Schematically, all release scenarios are shown in Fig. 1 (adapted from Fig. 1 of Toyota et al. (2011)). Herein, black arrows denote dry deposition of HOBr, BrNO$_3$, HBr, and O$_3$. Blue doted arrows indicate gas-phase photochemistry. The recycled

fluxes are displayed by dashed orange (O$_3$) and red (HOBr, BrNO$_3$, HBr) arrows.

## 2.2 Implementation

In accordance to the described scheme, submodel interface layer (SMIL), submodel core layer (SMCL), and namelist of ONEMIS have been extended based on EMAC version 2.52. Channel objects, used by a subroutine `airsnow_emissions`

(implemented in SMCL), include surface temperature (`tsurf`), fraction of snow cover on land (`cvs`), fraction of ice cover on ocean (`seaice`), cosine of sun's zenith angle (`cossza`), and dry deposition fluxes of HOBr, BrNO$_3$, HBr, and O$_3$ (`drydepflux_<HOBr, BrNO3, HBr, O3>`). Dry deposition is computed by submodel DDEP (formerly DRYDEP, Kerkweg et al., 2006a, b). In the SMIL of ONEMIS, these channel objects are defined and initialized and the subroutine

`airsnow_emissions` is called. Additional information about multi-year sea ice cover (MYSIC) has to be provided through data import. Currently, we are using a MYSIC estimate based on mean SIC from ERA-Interim (see Section 2.3). Steering parameters, $\Phi_1$, $T_{\rm crit}$, and $\theta_{\rm crit}$, can be changed in the corresponding control sequence within the ONEMIS namelist file. However, the parameter relevant to MY sea ice and LS in $\Phi_1$ is currently not used, since no parameterization has been provided

by Toyota et al. (2011). New output channels `snow_air_flux_br2` and `snow_air_flux_brcl` have been defined in the SMIL of ONEMIS. More detail of the algorithm implemented in the subroutine `airsnow_emissions` is provided as Nassi-Shneiderman diagram in Supplement S.2. The new emission mechanism has been named *AirSnow* and can be switched on in the ONEMIS namelist – an example excerpt has been added as Supplement S.3. After $Br_2$ has been released, we make use of EMAC's standard atmospheric bromine chemistry (Supplement S.1), that has been optimized for stratospheric condi-

tions (e.g., Sinnhuber and Meul, 2015; Falk et al., 2017). This treatment, however, might not be fully sufficient with respect to tropospheric heterogeneous chemistry on aerosols and should by subject to future work.

### 2.3 Validation Experiments

Four experiments have been performed using EMAC version 2.52 (see Table 1 for a summary). The basic model setup has been adapted from RC1SD-base-08, which is part of a Chemistry Climate Model Initiative (CCMI) recommended set of simulations

by the Earth System Chemistry integrated Modelling (ESCiMo) consortium (Jöckel et al., 2016). The model integrations use specified dynamics nudged to ERA-Interim for the year 2000. Accordingly, ERA-Interim sea ice cover (SIC) has been used. The chosen spatial resolution is T42L90MA corresponding to a $2.8° \times 2.8°$ grid, with a top level at $0.01\,\mathrm{hPa}$ and distributed to 90 levels. Output has been saved with 1-hourly temporal resolution. In contrast to RC1SD-base-08, fluxes of brominated very short-lived substances (VSLS), $CH_2Br_2$ and $CHBr_3$, are computed online from sea water concentrations (Ziska et al., 2013)

using the EMAC submodel AIRSEA (Pozzer et al., 2006) as described by Lennartz et al. (2015). In this scheme, sea ice acts as a lid blocking the emission of VSLS to the atmosphere. Comprehensive tropospheric and stratospheric chemistry as well as heterogeneous reactions within MECCA (Sander et al., 2011) have been activated for an aerosol surface area concentration climatology.

The basic parameter setup has been adopted without changes as proposed by Toyota et al. (2011). The temperature threshold

for all simulations has been $T_{\rm crit} = -15°\,\mathrm{C}$, accordingly.

In EMAC no discrimination is made between FY sea ice and MY sea ice, therefore we initially assume all ice to be first-year (BrXplo_fysic). A multi-year sea ice cover has been computed from RC1SD-base-08 10-hourly SIC output, which is based on ERA-Interim. We regard ice at a fixed location that survived one melting season as multi-year. Hence for simplicity, we assume no drift of ice masses. SIC has been integrated for respective summer months on northern (August/September) and

30 southern (February/March) hemisphere. The SIC at the minimum of the integrated SIC has been chosen as MYSIC for the respective year after. The resulting MYSIC for the year 2000 are shown in Fig. 2 together with monthly mean SIC for April (northern hemisphere) and September (southern hemisphere). The result is very similar with regard to patterns and extend of MYSIC on maps retrieved from satellite observation (US National Snow & Ice Data Center (NSIDC), 2017). Based on this MYSIC estimate, a second model integration (BrXplo_mysic) has been conducted. For comparison, a reference simulation

**Table 1.** EMAC model experiments used in this study. All experiments have been done using specified dynamics nudged to ERA-Interim. Accordingly, ERA-Interim SIC has been used. The setup is based on the consortial ESCiMo simulation RC1SD-base-08. Experiments have been performed for an assumption of first-year sea ice only (FYSIC) and for a multi-year sea ice cover (MYSIC) estimated from SIC. The temperature threshold for all simulations has been $T_{\mathrm{crit}} = -15\,^\circ\mathrm{C}$, accordingly.

| Experiment | Model Version | Resolution | Time-Span | Chemistry | VSLS Emission | AirSnow | $r_{O_3}^{\mathrm{ice-snow}}$ |
|---|---|---|---|---|---|---|---|
| BrXplo_ref | 2.52 | T42L90MA | Jan–Dec 2000 | full | AIRSEA | no | $2000\,\mathrm{sm}^{-1}$ |
| BrXplo_fysic | 2.52 | T42L90MA | Jan–Dec 2000 | full | AIRSEA | FYSIC | $2000\,\mathrm{sm}^{-1}$ |
| BrXplo_mysic | 2.52 | T42L90MA | Jan–Dec 2000 | full | AIRSEA | MYSIC | $2000\,\mathrm{sm}^{-1}$ |
| BrXplo_rs | 2.52 | T42L90MA | Jan–Dec 2000 | full | AIRSEA | MYSIC | $10000\,\mathrm{sm}^{-1}$ |

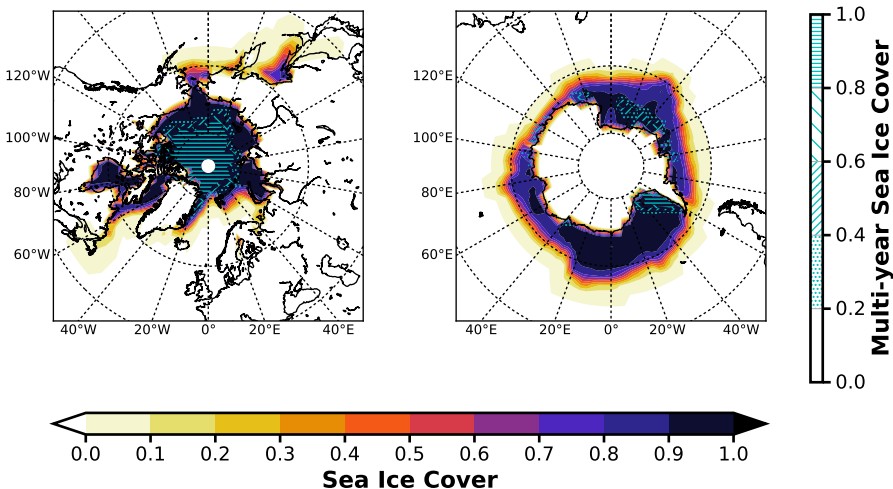

**Figure 2.** Sea ice cover fraction and estimated multi-year sea ice cover fraction for the year 2000. Mean SIC are shown for April in the northern hemisphere and September in the southern. MYSIC has been computed from RC1SD-base-08 10-hourly SIC based on ERA-Interim. For simplicity, we assume ice that survived one melting season at a fixed location as multi-year. (left) Northern hemisphere; (right) Southern hemisphere.

with bromine release mechanism switched off has been done (referred to as BrXplo_ref). In a further sensitivity simulation, we have decreased the dry deposition of ozone over snow covered regions as proposed by Helmig et al. (2007) by changing the surface resistance in DDEP for ozone on snow and ice surfaces from the value of $r_{O_3}^{\mathrm{ice-snow}} = 2000\,\mathrm{sm}^{-1}$ (Wesely, 1989) to $r_{O_3}^{\mathrm{ice-snow}} = 10000\,\mathrm{sm}^{-1}$ (Helmig et al., 2007).

## 3    Results

In this section, we compare the results of our model experiments with observational data. $Br_2$, which has been released from ice and snow, is transformed into $BrO$ photolytically. The enhancements of $Br_2$, therefore, lead to an increase of $BrO$ vertical column density. At first, we assess the spatial distribution of $BrO$ total VCD on global scales and have a brief look at the
temporal variations of $BrO$ VCD. Therefore, we compare EMAC (BrXplo_mysic) with GOME retrieved total VCD in both hemispheres (Sect. 3.1). Afterwards, we showcase implications on ODE regarding their temporal occurrence at specific ground-based observation sites in both hemispheres and discuss the limitations of the current bromine release mechanism (Sect. 3.2).

### 3.1    Total $BrO$ vertical column density

We first qualitatively discuss similarities and differences in spatial patterns between $BrO$ total VCD observation and model.
Afterwards, we assess the same patterns in a more quantitative way, comparing VCD anomalies for both, observation and model, with respect to monthly zonal averages.

In Fig. 3a), monthly mean $BrO$ VCD from GOME retrievals are shown for both, northern and southern, polar regions in April and September, respectively. In April, GOME data display enhanced $BrO$ VCD across the whole coastal region of the Arctic ocean down the Hudson Bay. There are signs of slight enhancements in the Antarctic coastal regions, where data are available,
but no hot spot can be determined. In September, enhancements above Antarctica are in particular found in the Ross and Weddell sea areas. As expected, no enhancements are found in the northern hemisphere.

From 1-hourly $BrO$ profiles of the EMAC model output, a total VCD has been integrated and re-sampled to 10–11 am local solar time, according to the ERS-2 equator crossing time of 10.30 am local time. Strictly speaking, this is not correct in general, for transition times at high latitudes differ from the equator crossing time due to the satellite orbit. Differences in local time may
account for part of the differences seen in the following $BrO$ comparison. The re-sampled data has been averaged monthly and are shown in Fig. 3b). Note, as there is an offset between EMAC and GOME $BrO$ VCD (roughly $(2-4) \cdot 10^{13}$ molecules $cm^{-2}$ less in case of EMAC), we have chosen different color scales for each dataset to illustrate the spatial similarities rather than the difference in the absolute VCD. Qualitatively, the spatial patterns of $BrO$ VCD are reasonably well reproduced by EMAC. In the northern hemisphere in April, $BrO$ VCD are relatively overestimated westward from Novaya Zemlya to the east coast
of Greenland compared to GOME observations. In September, there is no significant enhancement, which is in accordance to observations. Regarding the Southern hemisphere in September, $BrO$ VCD spatial patterns are quite similar, displaying main $BrO$ enhancements in the Ross and Weddell sea areas, while in April we identify a hot spot in the Ross sea area that is not shown by the observations.

To highlight the BE events, we have computed anomalies of monthly mean VCD of $BrO$ for both, GOME and EMAC, by
subtracting the corresponding zonal means (Fig. 4). The associated zonal means are available as Supplement S.4. This allows for a more detailed assessment of the spatial patterns. In April, GOME data display a strong enhancement of $BrO$ VCD across the whole coastal region of the Arctic ocean, except for the northern coast of Greenland. The largest hot spot is found down the Hudson Bay, and minor hot spots appear east of Novaya Zemlya in the Sea of Ochotsk, and Hokkaido. A corresponding

negative anomaly is located across the Barents, Greenland and Norwegian sea. There are only slight enhancements in the Antarctic coastal regions. In September, hot spots around Antarctica can be in particular observed in the Ross and Weddell sea areas with corresponding negative anomalies located across the Bellingshausen and Amundsen sea.

In comparison to GOME data, spatial patterns of BrO VCD are astonishingly well reproduced by EMAC. Only westward from
the Hudson Bay no BrO enhancement is found and there is no negative anomaly between Greenland and Novaya Zemlia in the model. The Hokkaido hot spot appears slightly shifted northward. In September, both observation and model, agree well in both hemispheres. The hot spot in the Ross sea area that occurs in the simulation cannot be identified from satellite observations for these are rather sparse in April. One needs to be cautious when drawing conclusions solely based on these comparisons, for we have used total VCD the actual BE events might as well be disguised by variation in the stratospheric BrO column.
A full overview of monthly mean total BrO VCD for both, observation and model, including all months has been included as Appendix A (Figs. A1-A4), respectively. In the northern hemisphere, the implemented mechanism is apparently prone for BrO VCD enhancements shifted to early winter compared to GOME retrievals. The occurrence of BE events in fall is not supported by any observation. In late spring and early summer, however, too few BrO is formed in the model. This may hint to sources of BrO in the Arctic, which are not represented by this mechanism or an adherence to the chosen parameters. Further studies
would be needed to resolve the source of this discrepancy. In the southern hemisphere, the modeled BrO enhancements in, e.g., August and September are similar in their occurrence, while the sparseness of GOME data in austral winter does not permit further conclusions regarding the quality of the parameterization in this region. Taking a look at the zonally averaged total BrO VCD (Supplement S.4), we find, that the modeled BrO VCD is generally too small in polar summer compared to observation by about $(1-4) \cdot 10^{13}$ molecules cm$^{-2}$ in both hemispheres, respectively. A better agreement between observation and model
is achieved in winter. This is due to the implementation of the bromine release mechanism (doted lines indicating the reference simulation). Hence, taking the bromine released from ice and snow into account the overall model performance is enhanced with respect to polar BrO observation.

## 3.2   Ozone depletion events

Regarding depletion events of surface ozone, four different observation sites have been chosen in each hemisphere for com-
parison (Table 2). No data for Arrival Heights and Palmer Station have been available in 2000. For these stations, we show model results only. Time series of surface ozone VMR are shown in Figures 5–6 including both in situ observations (where available) and model simulations. For each simulation, the nearest grid point has been chosen as representative. In general, we find a good agreement between BrXplo_ref and observations for seasons without bromine release from ice and snow, except for Summit, South Pole station, and Neumayer station in austral winter, where model results are systematically lower compared to
observations. In case of BrXplo_fysic, all northern hemispheric sites display depletion events in spring as well as in fall. While the depletion events are not entirely in temporal coincidence with observed events, their frequency is generally well reproduced. However, events of ozone depletion in fall are not present in observation data. For Zeppelin Mountain and Alert, these *fault events* are due to the FYSIC assumption. For a decent multi-year sea ice cover is implemented in BrXplo_mysic, most of them vanish. In case of Barrow, a closer look into spring reveals an astonishing temporal as well as quantitative coincidence of

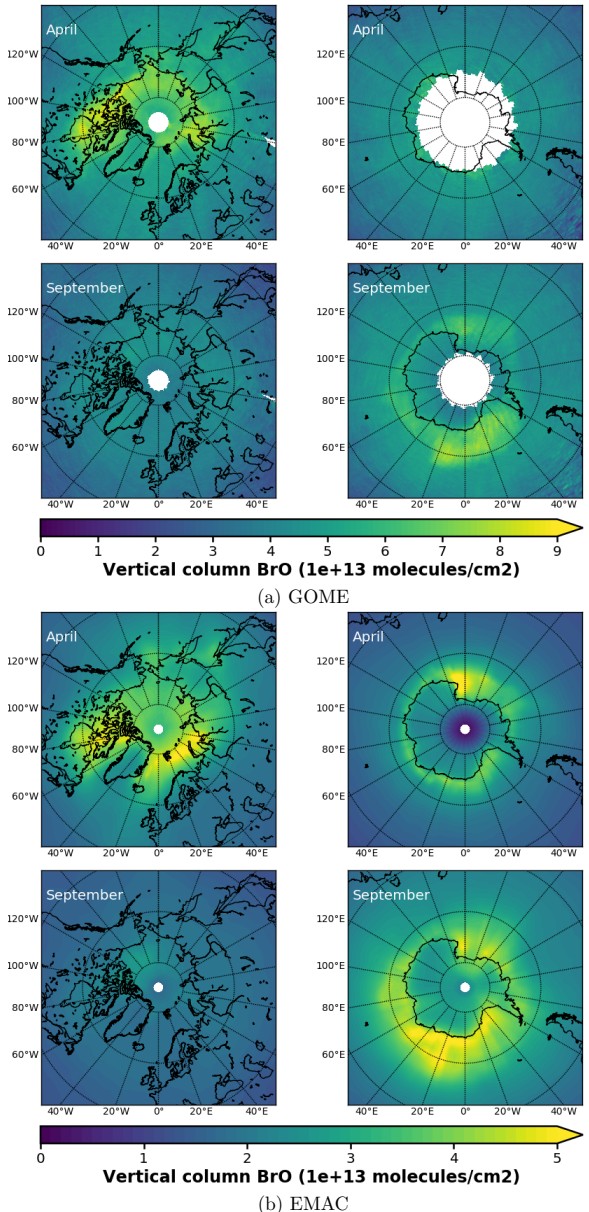

**Figure 3.** Monthly mean total VCD of BrO for the Arctic and Antarctic (April and September), respectively. EMAC data have been sampled in accordance to local solar time 10–11 am. Different color scale ranges have been chosen to illustrate the similarities in the spatial distribution of the data rather than the absolute amount of BrO VCD. (a) GOME; (b) EMAC (BrXplo_mysic).

surface ozone VMR especially in April (Fig. 6). The apparent *wiggles* are partly due to hard trigger thresholds $T_{\mathrm{crit}}$ and $\theta_{\mathrm{crit}}$, but similar structures are in fact apparent in the surface ozone observations at Barrow implying a diurnal variation of $O_3$. At

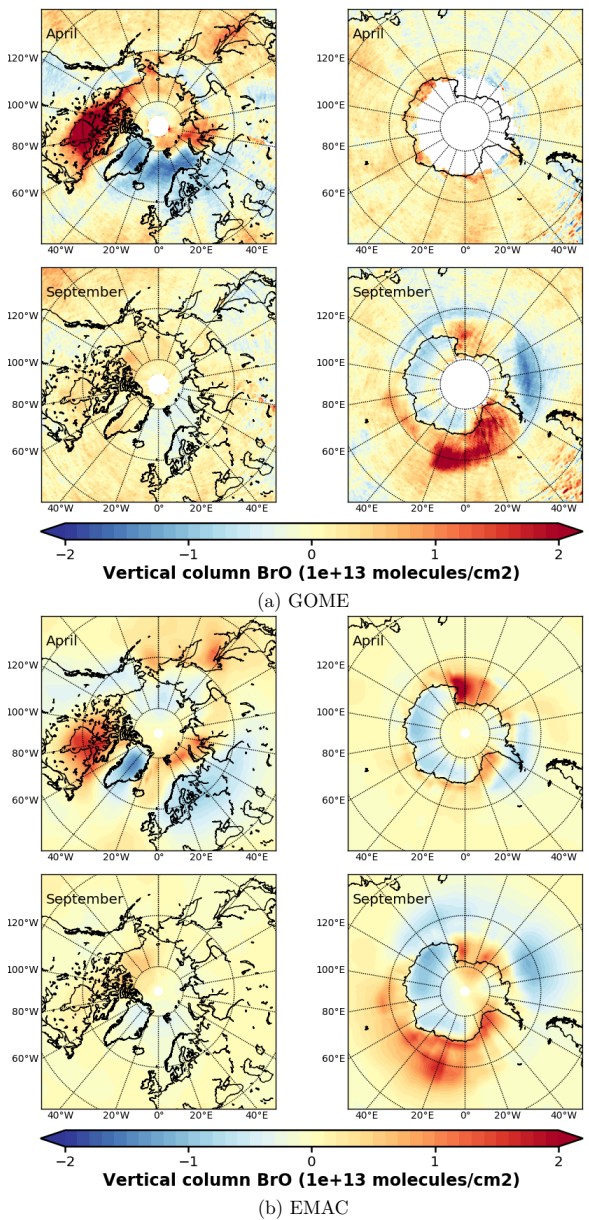

**Figure 4.** Anomalies of monthly mean VCD of BrO for the Arctic and Antarctic (April and September) with respect to monthly averaged zonal mean (see Supplement S.4), respectively. EMAC data have been sampled in accordance to local solar time 10–11 am. (a) GOME; (b) EMAC (BrXplo_mysic).

Alert, our model does not capture the 2000s ODE that inflicted continuously low surface ozone levels for several days from late April until early May. As pointed out by Strong et al. (2002), this long-lasting depletion event was related to transport of ozone

**Table 2.** Observation sites for surface ozone comparison. However, for Palmer station and Arrival Heights no observations of surface ozone are available for the year 2000, so that we present model results only for these two stations.

| Site | Location | Latitude (°N) | Longitude (°E) | Altitude (m a.s.l.) | Data Provider |
|------|----------|---------|-----------|----------|---------------|
| Alert | Canada | 82.50 | -62.30 | 210 | EBAS (NILU) |
| Barrow | Alaska | 71.32 | -156.61 | 8 | ESRL/GMD (NOAA) |
| Zeppelin Mountain | Spitsbergen | 78.90 | 11.88 | 474 | EBAS (NILU) |
| Summit | Greenland | 72.54 | -38.48 | 3238 | ESRL/GMD (NOAA) |
| Palmer Station | Antarctica | -64.77 | -64.05 | 21 | *ESRL/GMD (NOAA)* |
| Neumayer Station | Antarctica | -70.68 | -8.26 | 43 | EBAS (NILU) |
| Arrival Heights | Antarctica | -77.85 | 166.78 | 22 | *ESRL/GMD (NOAA)* |
| South Pole Station | Antarctica | -89.98 | -24.8 | 2810 | ESRL/GMD (NOAA) |

poor air originating from a region north of Ellesmere Island and the eastern arctic ocean, respectively. It is not clear whether transport of ozone depleted air masses or depletion itself is too weak in our simulation. At about the same time (late April, early May) observation displays a series of ODE at Zeppelin mountain, which is also only partly reproduced by the model (e.g. on April 28th). Comparing observation and simulation in the southern hemisphere and Greenland, we find in general less
ozone in BrXplo_mysic as well as in BrXplo_ref. Reducing the ozone dry deposition over snow and ice (BrXplo_rs) slightly increases boundary layer ozone at all discussed sites (see Supplement S.5). But even with this reduced dry deposition, the model significantly underestimates observed boundary layer ozone in Antarctica, indicating that there are missing sources of ozone release from ice and snow in the model (e.g., Oltmans, 1981; Helmig et al., 2007). Any analysis regarding the modeled occurrence of ODE in the southern hemisphere is not affected by this. Despite the original mechanism's validation for northern
hemispheric spring (Toyota et al., 2011), comparison of time series for the southern hemisphere do display ozone depletion events in a similar frequency as found in observational data. At Neumayer station, we find some events in late October and early November that might be coincidental, but in most cases simulated ODE show up later than actually observed ODE.

In summary, while some aspects of ODE are reproduced remarkably well by the implemented mechanism, especially the long-lasting event at Alert is not reproduced at all. This strongly hints to the involvement of further mechanisms, e.g., blowing snow
and sea spray or even anthropogenic $NO_x$ (Custard et al., 2015), in the depletion of polar surface ozone which have not yet been modeled in EMAC. In BrXplo_rs with reduced dry deposition, ozone depletion events in fall and midwinter are suppressed and the agreement with observed ozone is generally improved. Following Toyota et al. (2011, see Fig. 10), who have investigated how well the proposed mechanism reproduces individual bromine explosion events by correlating daily tropospheric GOME BrO VCD with modeled BrO, we assess the correlation between observed and modeled surface ozone. The correlation be-
tween observed and modeled surface ozone at Barrow is shown in Fig. 7. Additional correlation plots for the other stations in the northern hemisphere, as well as additional plots for the sensitivity simulation with reduced ozone dry deposition are pro-

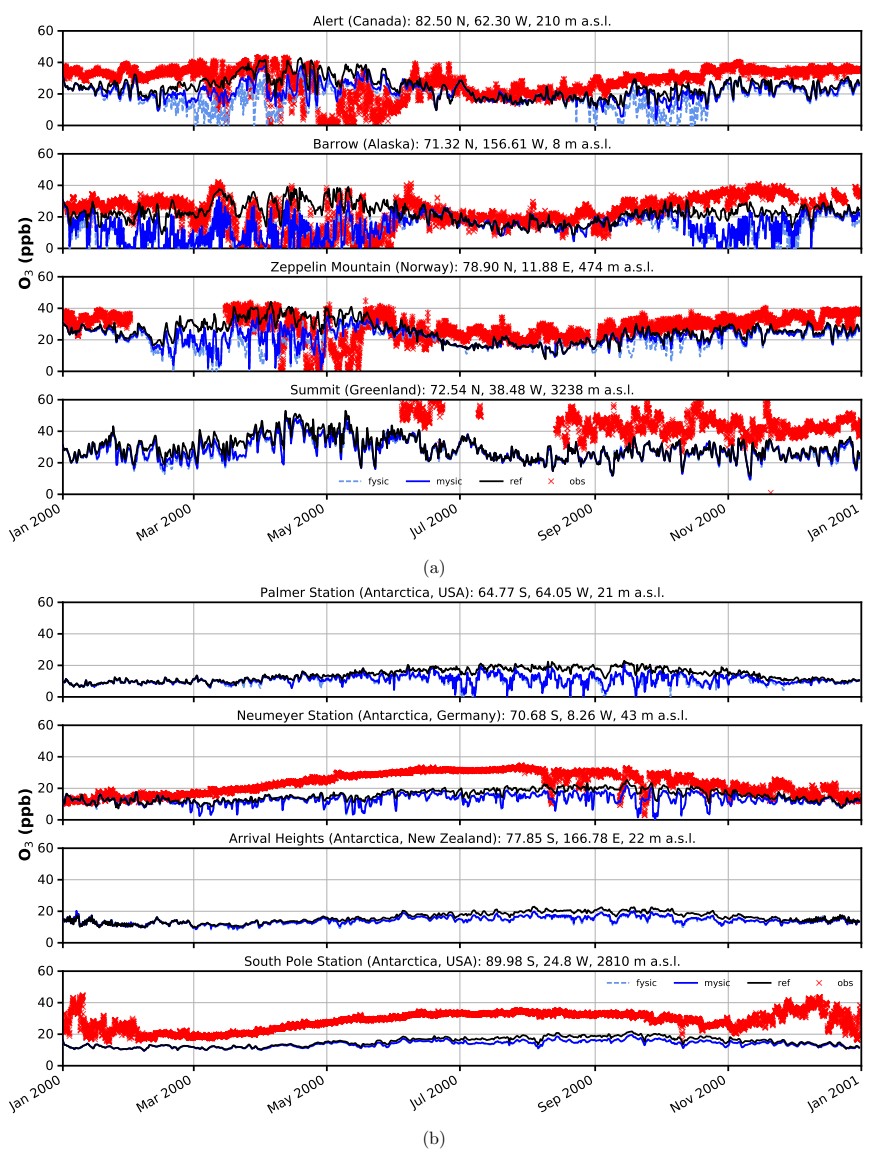

**Figure 5.** Surface ozone mixing ratios at four different observation sites. Comparison of in situ measurements (red crosses) with results from simulation (solid black – EMAC v2.52 default (no bromine explosions); light blue dashed – FYSIC; solid blue – MYSIC). Representatively, the nearest grid point has been chosen. (a) Northern hemisphere; (b) Southern hemisphere.

vided in the supplementary (S.5-S.6). As already evident from the time series in Fig. 6, low surface ozone values largely absent in the reference simulation are reproduced by the EMAC simulations with included bromine release mechanism. However, some *fault events* are also generated, which are not present in observations. Overall, the linear correlation coefficients between modeled and observed ozone are improved by inclusion of the bromine explosion mechanism (from 0.008 to 0.21 at Barrow).

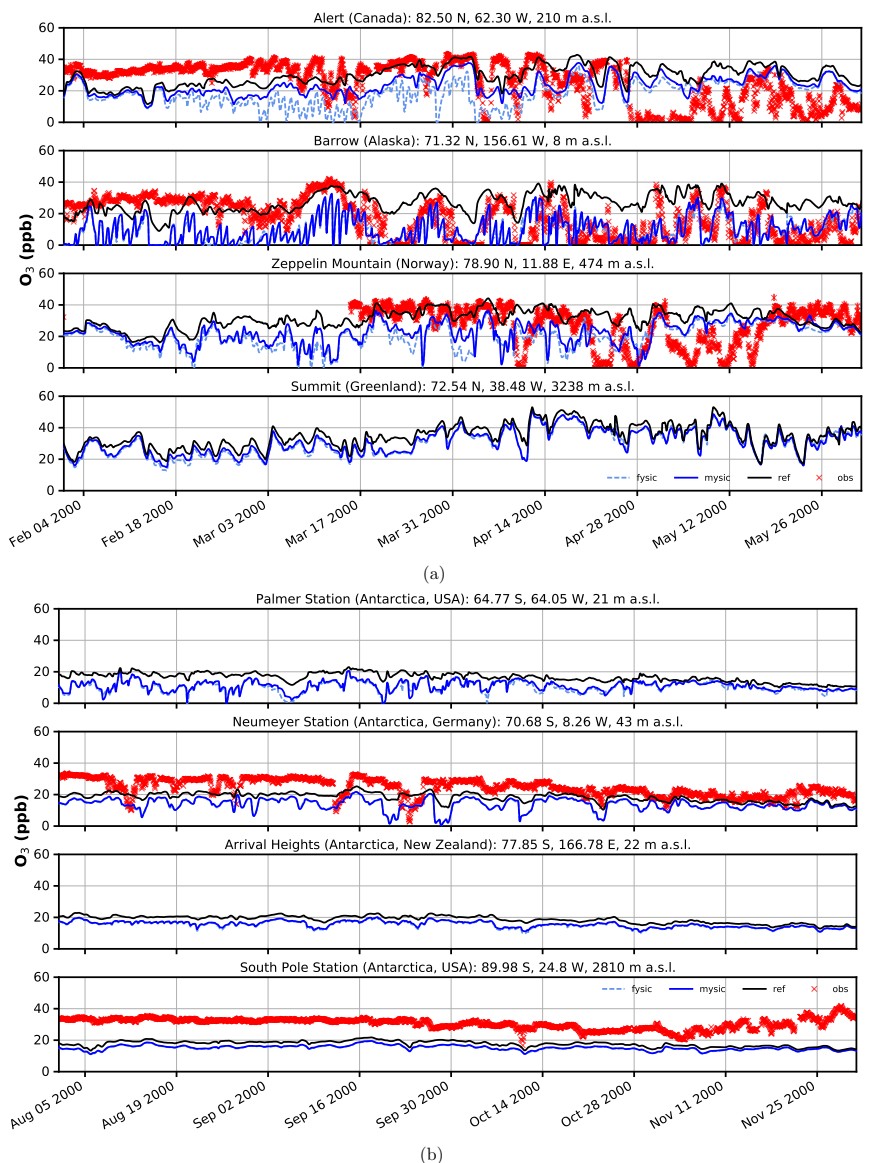

**Figure 6.** Surface ozone mixing ratios at four different observation sites for spring and austral spring, respectively. Comparison of in situ measurements (red crosses) with results from simulation (solid black – EMAC v2.52 default (no bromine explosions); light blue dashed – FYSIC; solid blue – MYSIC). Representatively, the nearest grid point has been chosen. (a) Northern hemisphere; (b) Southern hemisphere.

In Fig. 8, the time-lagged correlation coefficients between observed surface ozone and different model experiments (BrXplo_ref, BrXplo_mysic, and BrXplo_rs) are shown exemplarily for Alert and Barrow, respectively. Model data have been shifted with respect to the observation. Therefore a positive lag indicates a later occurrence of low surface ozone in the model

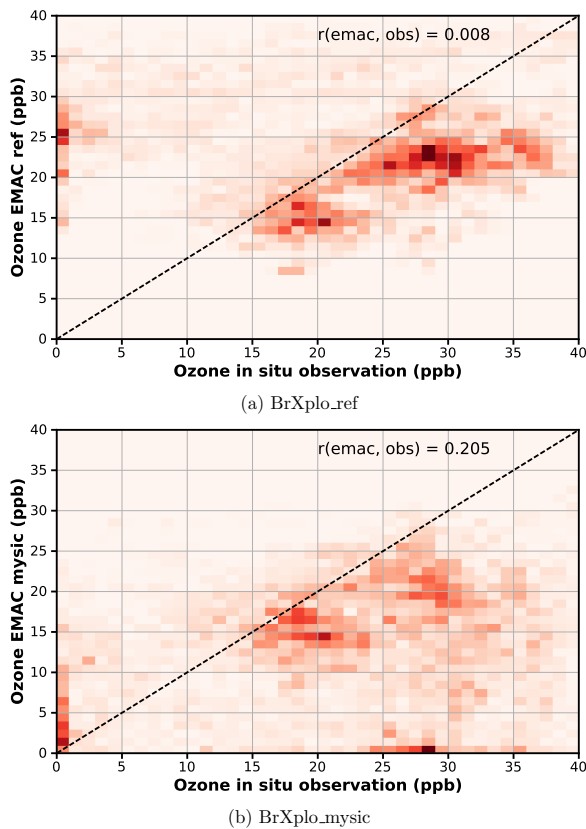

**Figure 7.** Temporal correlation of modeled surface $O_3$ with observation at Barrow. Data have been binned in bins of $1\,\mathrm{ppb}$ width. While observed low ozone events at Barrow are absent in the reference simulation, in the BrXplo_mysic simulation there is now a population where both observation and model simultaneously show low ozone values, which is also reflected in the improved linear correlation coefficient from 0.008 to 0.21. (a) BrXplo_ref; (b) BrXplo_mysic.

experiment. In case of Barrow, the previously diagnosed improvement of the correlation is further stressed. The time-lagged correlation peaks at a lag of about $2\,\mathrm{h}$, with correlations falling to half of the maximum at about $\pm 2$ days. Updating the ozone surface resistance over ice ($r_{\mathrm{O_3}}^{\mathrm{ice-snow}} = 10000\,\mathrm{s\,m^{-1}}$) further increases the correlation. In case of Alert, the correlation peaks between $-40$ and $-48\,\mathrm{h}$ regardless, which means low ozone values may occur about 2 days ahead of time in all model experiments. There is no improvement by switching on the bromine release mechanism. We conclude, that the mechanism transmuting HBr, HOBr, and $BrNO_3$ to $Br_2$ sufficiently parameterizes the main traits leading to the depletion of surface ozone in the Barrow region, but does not describe the situation at Alert well.

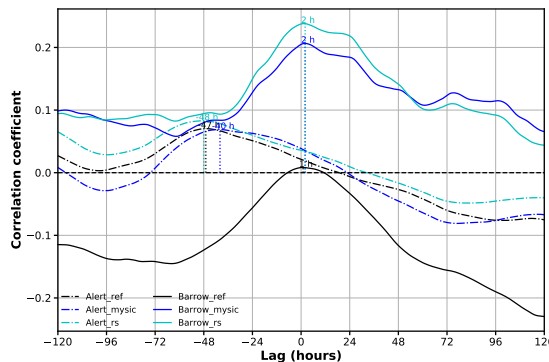

**Figure 8.** Time-lagged correlation coefficients between observed surface ozone and different model experiments (BrXplo_ref, BrX-plo_mysic, and BrXplo_rs) for Alert and Barrow, respectively. Model data have been shifted with respect to the observation. Therefore a positive lag indicates a later occurrence of low surface ozone in the model experiment.

## 4  Discussion and conclusions

We have implemented a bromine release mechanism from sea ice and snow covered land surfaces based on the parameteriza-tion suggested by Toyota et al. (2011) in the global chemistry-climate model EMAC. While in the original study Toyota et al. (2011) focused on Arctic spring time only, we extend the simulations to both hemispheres and a full annual cycle. Without

any further tuning of the parameters ($T_{\mathrm{crit}}$, $\theta_{\mathrm{crit}}$, and $\Phi_1$), our model simulations with this relatively simple mechanism suc-cessfully reproduce many observed features of bromine enhancement and ODE (spatially as well as temporally). The overall model performance regarding $\mathrm{BrO}$ VCD and surface ozone concentrations at high latitudes is improved.

The resulting spatial patterns of $\mathrm{BrO}$ total VCD are in good agreement with $\mathrm{BrO}$ VCD retrieval of the GOME satellite instru-ment. But one has to treat this warily, for in comparing total VCD the actual bromine explosion events might be disguised by

overlaying stratospheric $\mathrm{BrO}$ variations. Hence in a next step, tropospheric VCD should be computed using similar algorithms (e.g., Richter et al., 1998; Theys et al., 2011) on observational and simulation data, respectively. Despite these improvements, modeled $\mathrm{BrO}$ VCD is still generally underestimated in comparison to GOME data. In particular, an observed lag of $\mathrm{BrO}$ during respective summer and fall in both hemispheres is pointing to further missing sources of $\mathrm{BrO}$ in the model. Using satellite data, no firm conclusion can be drawn regarding the temporal occurrence of bromine explosion events. We have instead studied the

temporal occurrence of ODE comparing model data and in situ observation at different sites in both, the Arctic and Antarctic, respectively. While ODE are very well reproduced in case of Barrow, there are notable discrepancies at other observation sites. At Barrow, the time-lagged correlation coefficient analysis, which peaks at about zero lag between observation and model data, displays a significant enhancement if bromine explosions are taken into consideration. However, the improvement to reproduce individual ODE is less clear at other places. In particular at Alert, the model seems to generate ODE about 2 days ahead of

time. The recognized ODE, which had been observed at Alert in late April/early May in 2000, is not at all reproduced by

this bulk-snow-based mechanism. In general, there is a tendency to generate too many ODE in fall and mid-winter, which is reduced by decreasing the ozone dry deposition ($r_{O_3}^{\mathrm{ice-snow}}$). Using a reasonable multi-year sea ice cover estimate also reduces the occurrence of *fault events* in fall.

The implemented bromine release mechanism relies on various assumptions, e.g., $T_{\mathrm{crit}}$, $\Phi_1$. Though these have been cross-validated with observations by Toyota et al. (2011), they are not entirely constrained. The chosen temperature threshold might be too low regarding the actual physical processes. Dynamical factors such as wind speed increasing $Br_2$ release through pumping or ventilation of the snow are entirely neglected in this parameterization. In addition, the dry deposition, which is one of the key factors in this bromine release mechanism, is still highly uncertain and hard to measure explicitly. Since heterogeneous chemistry on aerosols in the polar boundary layer plays an important role, this is a topic which needs to be elaborated on.

It is plausible, that in reality different processes, such as bromine activation by blowing snow, sea spray, or even by $NO_x$ from unaccounted anthropogenic sources (Custard et al., 2015), all play a role and contribute to the bromine explosion events at different sites. Assuming blowing snow as source of bromine enriched sea salt aerosols, Yang et al. (2010) and Theys et al. (2011) have shown that many bromine explosion events are reproduced in duration, location, and magnitude for Antarctic sites. But they remain rather vague in their assessment of discrepancies between model and observation. Comparing both schemes within the same model environment could help to gain a better understanding of bromine explosion events and subsequent ozone depletion from the modeling perspective. As shown in this work, MESSy provides a framework, in which the various bromine explosion schemes can be implemented relatively straight forward.

With the implemented scheme, following Toyota et al. (2011), and the corresponding model experiments, we have now a basis for, e.g., the validation of bromine explosion events at specific sites using in situ and ground-based $BrO$ data, the evaluation of modeled temporal correlation between BE events and ODE, the validation of usage of online aerosol formation in the polar boundary layer, the validation of heterogeneous chemistry in the polar boundary layer, the implementation and validation of a blowing snow scheme with respect to observation, and the comparison of these two mechanisms.

*Code availability.* The Modular Earth Submodel System (MESSy) is continuously further developed and applied by a consortium of institutions. The usage of MESSy and access to the source code is licensed to all affiliates of institutions, which are members of the MESSy Consortium. Institutions can become a member of the MESSy Consortium by signing the MESSy Memorandum of Understanding. More information can be found on the MESSy Consortium Web-site (http://www.messy-interface.org). The modified code of the submodel ONEMIS described here will be made available with the next official release of the MESSy source code distribution.

*Data availability.* For any party interested, model results can be made available on request.

# Appendix A: Total $BrO$ vertical column density

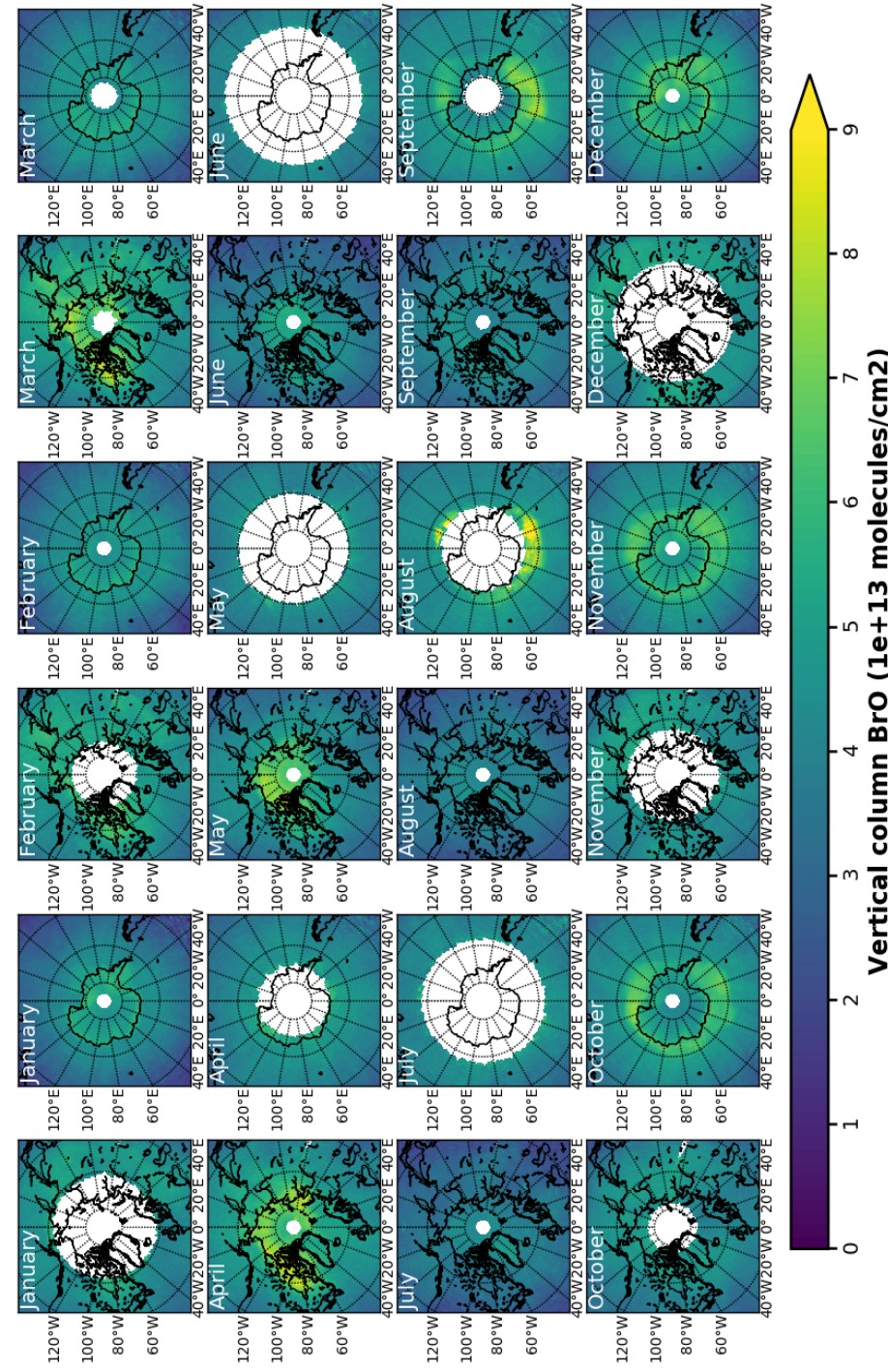

**Figure A1.** GOME monthly mean total VCD of BrO for the Arctic and Antarctic.

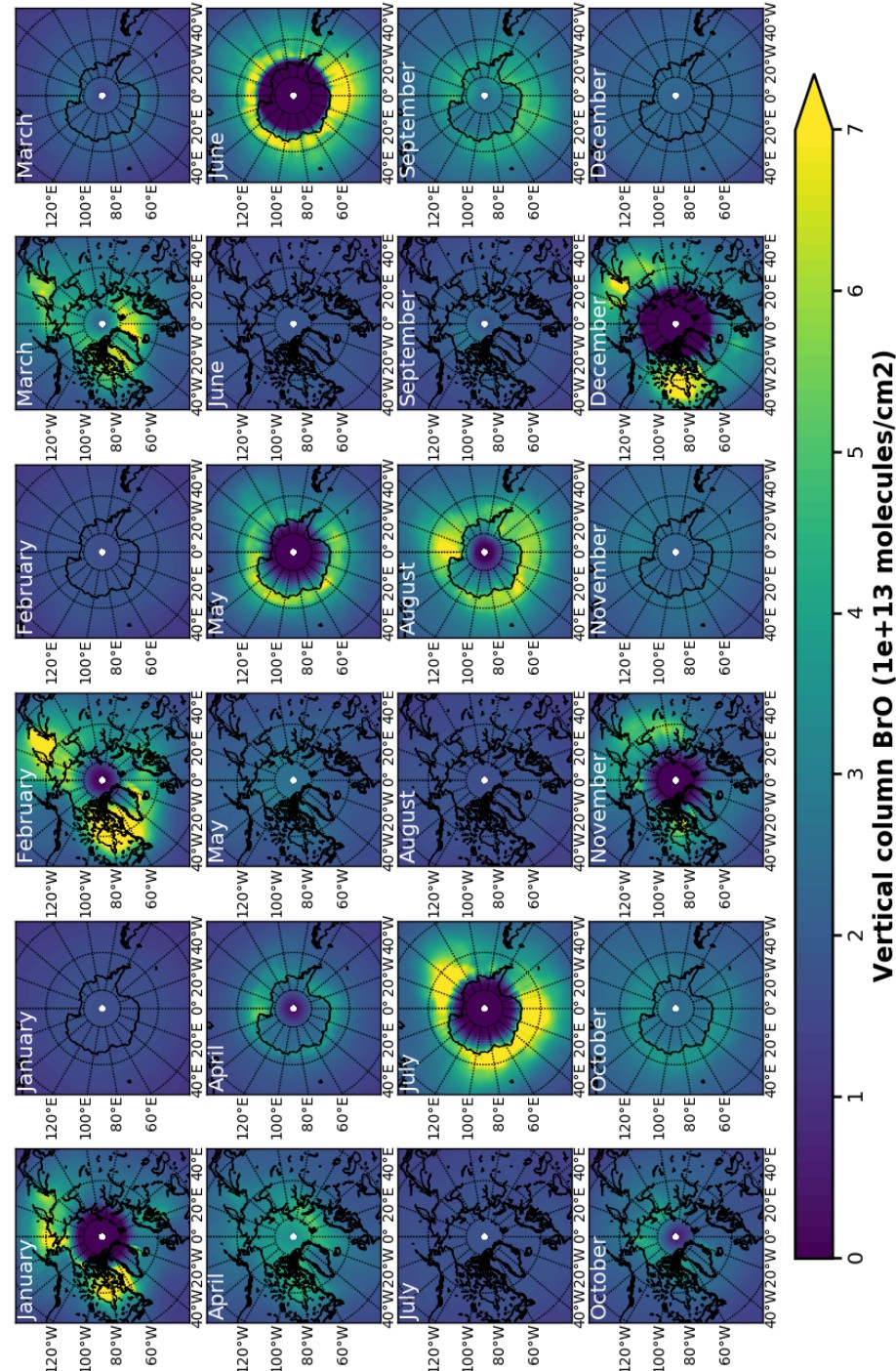

**Figure A2.** EMAC (BrXplo_mysic) monthly mean total VCD of BrO for the Arctic and Antarctic. EMAC data have been sampled in accordance to local solar time 10–11 am.

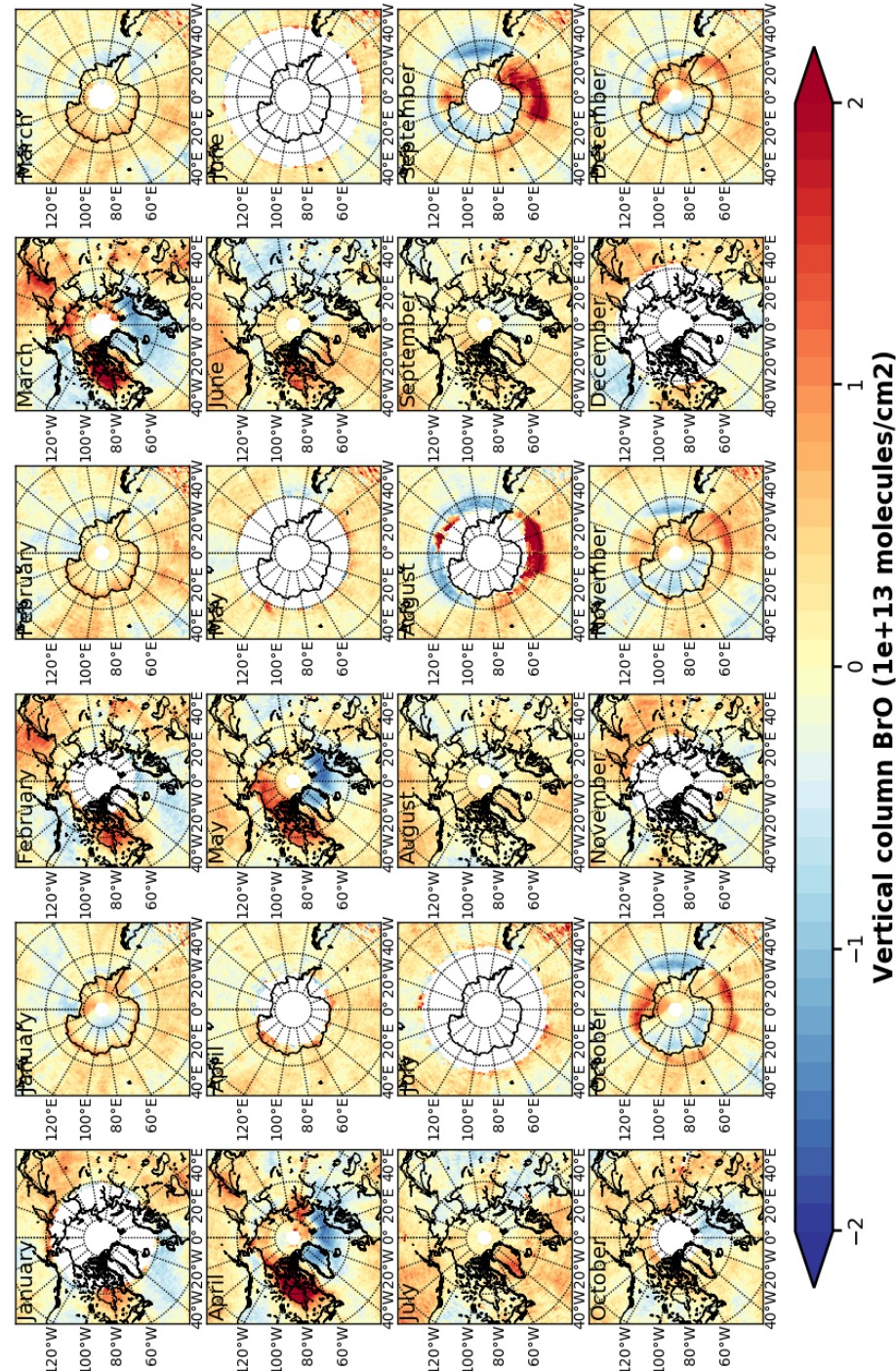

**Figure A3.** Anomalies of GOME monthly mean VCD of BrO for the Arctic and Antarctic with respect to monthly averaged zonal mean (see Supplement S.4).

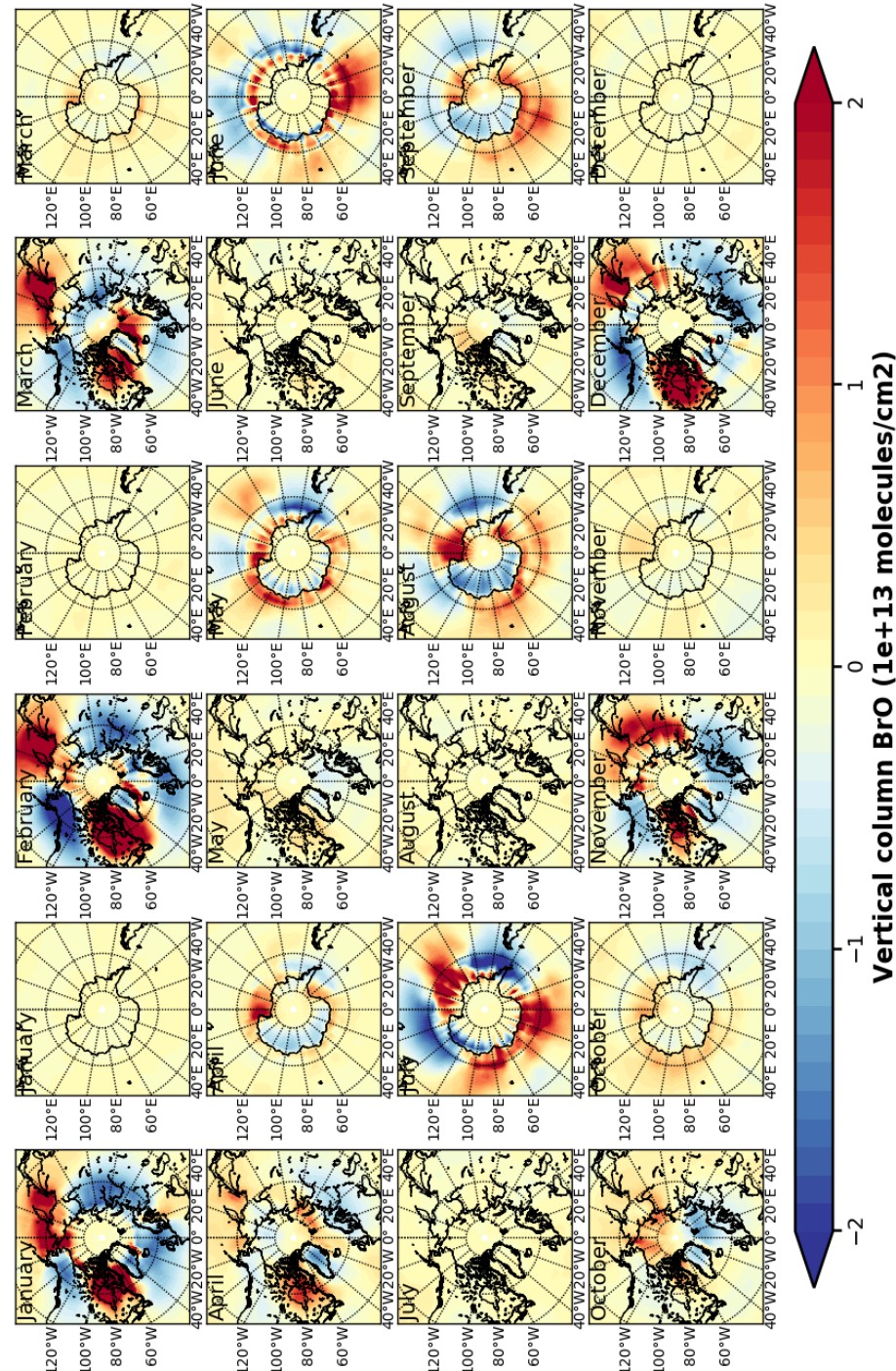

**Figure A4.** Anomalies of EMAC (BrXplo_mysic) monthly mean VCD of BrO for the Arctic and Antarctic with respect to monthly averaged zonal mean (see Supplement S.4). EMAC data have been sampled in accordance to local solar time 10–11 am.

*Author contributions.* Stefanie Falk has implemented the described mechanism, run and validated the simulations with observational data. Björn-Martin Sinnhuber suggested this study and took part in the analysis. Both authors contributed to the writing of the paper.

*Competing interests.* The authors declare that they have no conflict of interest.

*Acknowledgements.* Parts of this work were supported by the Deutsche Forschungsgemeinschaft (DFG) through the research unit 'SHARP'
(SI1044/1-2), the German Bundesministerium für Bildung und Forschung (BMBF) through the project 'ROMIC-THREAT' (01GL1217B), and by the Helmholtz Association through its research program 'ATMO'.

Ozone in situ data for Alert, Neumayer station, and Zeppelin Mountain have been made available by the Norwegian Institute for Air Research. Database of observation data of atmospheric chemical composition and physical properties, EBAS. http://ebas.nilu.no. Data of Alert are provided by Environment Canada / Atmospheric Environmental Service (EC/AES), data of Neumayer station by Helmholtz-Zentrum
Geesthacht (HZG), and data of Zeppelin Mountain by Norwegian Institute for Air Research (NILU).

Ozone in situ data for Barrow, Summit, and South Pole station have been provided by U.S. Department of Commerce/National Oceanic & Atmospheric Administration (NOAA) – Earth System Research Laboratory – Global Monitoring Division. https://www.esrl.noaa.gov/gmd/ozwv/surfoz.

Tropospheric BrO column retrievals from GOME instrument have been provided in courtesy of Andreas Richter and John P. Burrows (Uni-
15 versity of Bremen). The data can be obtained from http://www.iup.uni-bremen.de/doas/gome_bro_data.htm.

We thank Andreas Richter and Astrid Kerkweg for helpful comments on an earlier version of the manuscript.

Thanks to Stefan Versick (KIT SimLab Climate and Environment) for technical support concerning the implementation into the EMAC model.

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
