# Peer review of "Polar boundary layer bromine explosion and ozone depletion events in the chemistry-climate model EMAC v2.52: Implementation and evaluation of AirSnow algorithm"

_Geoscientific Model Development, 2017_

## Referee Comment (RC1) · Anonymous Referee #1 · 2 Aug 2017

Comments on Falk and Sinnhuber manuscript titled 'Polar boundary layer bromine explosion and ozone depletion events in the chemistry-climate model EMAC v2.5.2: Implementation and evaluation of AirSnow algorithm'

**General comments:**

This work reports an implementation of a simple treatment of bromine release and cycling on sea ice and snow in a global chemistry-climate model ECHAM-EMAC, following the work of Toyota et al. (2011). Then the author and co-author run the model for a whole year (2000) to validate the model by comparison with satellite GOME-SLIMACAT tropospheric BrO VCD and surface ozone observation at some selected sites in both the northern and the southern hemispheres. What they concluded is that the Toyota et al scheme of bromine release from snow works in general, as the model could reproduce the spatial patterns of BrO VCD as shown in the satellite data, and simulate surface ozone depletion events. They also pointed out some discrepancies between model and obs. For example, most of simulated ODEs are seen in the northern hemisphere, rather in the south.

However, I find that the validation of the model is too simple, though it mainly follows the method applied in Toyota et al. (2011). For example, they only compare model's monthly mean BrO VCD (in April and September) with satellite data. They did show any TEMPORAL comparison between them. I would suggest the author and co-author select some sites, such as those for ozone comparisons in figure 4-5 (Alert, Barrow, Summit, Palmer, Arrival Height and the southern Pole), and get corresponding model BrO VCD according to the satellite overpass time to give a scatter plot basing on the whole year data, as did in Yang et al. (2010). This kind of comparison could allow us examine the TIMING of the bromine release mechanism, though the time resolution will not better than ~24hrs. To allow a better comparison, a lead-lag (or lagged) relationship, e.g. by 1-2 days, of the model BrO can be used. By doing so, we could have a better understanding of the mechanism proposed/applied in the model.

My second major concern is the missing of sea spray acting as a bromine source to the troposphere in the modelling. The release of bromine from sea salt aerosols has been parameterised in various global chemistry models (e.g. Yang et al. 2005; Breider et al., 2009; Paralla et al, 2012). Why this kind of source is not included in the EMAC? Maybe this part is out of the topic of the manuscript (on air-snow emission), but a discussion covering this issue should be given.

For the above reasons, I would suggest a major revision to this current version before I suggest consideration of publishing it on GMD.

**Specific comments:**

P2L1: a review paper by Abbatt et al. 2012 should be cited here.

P2L22: removal the pair of bracket in '(boundary)'

P2 L23: why italic 'online' is used here?

P3 L 20-21: Some discussions should be given to explain why such as a higher value (7.5%) of molar yield at solar zenith angle >85° (comparing to 0.1% at dark) is introduced in the model, though this number is from Toyota et al paper. This parameter is one critical parameter to allow enough bromine releasing from snow to match the observation. Either a justification, e.g. reference, or a caution must be given to remind readers of what is going on here.

P6 figure 2 and P7L1-2: is the EMAC BrO VCD shown here a total of tropospheric and stratospheric BrO? If so, then a tropospheric column value should be worked out to make a direct comparison with satellite-based tropospheric BrO.

P7L1-9: as mentioned in the general comment, just a spatial comparison for BrO is not good enough, a temporal comparison between daily satellite BrO VCD and corresponding model BrO should be given here to allow a further examination of the bromine releasing mechanism applied.

Abbatt, T. P. D.; Thomas, J. L.; Abrahamsson, K.; Boxxe, C.; Granfors, A.; Jones, A. E.; King, MD; Saiz-Lopez, A.; Shepson, P. B.; Sodeau, J.; Toohey, DW; Toubin, C.; von Glasow, R.; Wren, S. N.; Yang, X.; Halogen activation via interactions with environmental ice and snow in the polar lower troposphere and other regions. *Atmospheric Chemistry And Physics*, **12**, 6237-6271, [doi:10.5194/acp-12-6237-2012] 2012.

Breider, T. J., M. P. Chipperfield, N. A. D. Richards, K. S. Carslaw, G. W. Mann, and D. V. Spracklen (2010), Impact of BrO on dimethylsulfide in the remote marine boundary layer, Geophys. Res. Lett., 37, L02807, doi:10.1029/2009GL040868.

Parrella, J. P., Jacob, D. J., Liang, Q., Zhang, Y., Mickley, L. J., Miller, B., Evans, M. J., Yang, X., Pyle, J. A., Theys, N., and Van Roozendael, M.: Tropospheric bromine chemistry: implications for present and pre-industrial ozone and mercury, Atmos. Chem. Phys., 12, 6723-6740, https://doi.org/10.5194/acp-12-6723-2012, 2012.

Yang, X., R. A. Cox, N. J. Warwick, J. A. Pyle, G. D. Carver, F. M. O'Connor, and N. H. Savage (2005), Tropospheric bromine chemistry and its impacts on ozone: A model study, J. Geophys. Res., 110, D23311, doi:10.1029/2005JD006244.

---

## Referee Comment (RC2) · Anonymous Referee #2 · 4 Aug 2017

This paper presents a new implementation of the mechanism of halogen activation presented in Toyota et al., 2011 in the model EMAC (ECHAM/MESSy Atmospheric Chemistry). I am supportive of this work being eventually published in GMD as it is well within the scope of the journal and presents a new model tool that can be used by the community to understand halogen activation and its role in ozone depletion events (ODEs). However, there are a number of issues that need to be addressed before publication. I largely agree with the first reviewer's comments and have in addition

some major and minor comments, detailed below, that should be addressed prior to publication.

Major comments:

1. How is bromine recycling on aerosol treated? Is this important to sustain halogen activation and does it contribute to ozone depletion events?

2. The authors should look deeper into the literature as to how the understanding of halogen chemistry in the Arctic & Antarctic has developed over time. Papers such as Barrie et al., 1988 and Abbatt et al., 2012 should not be omitted from the reference list. In addition, Simpson et al., 2007 provides an excellent overview of how our understanding of halogen chemistry and ODEs has developed.

3. A clearer discussion of how snow contributes to halogen activation is needed, as discussed by Pratt et al., 2013 and Thomas et al., 2011.

4. A list of the reactions that are included to describe the halogen cycle is needed either in the paper or in the supplement, including a short discussion of how heterogeneous reactions on aerosols are treated.

5. In general, I find the discussion of the results too short. Major features of the figures are not really described, which leaves the reader a bit lost as to what the model validation section means. For example, why is the surface ozone so low in the model compared to the measurement sites in Antarctica (Neumeyer and South Pole Stations)? If the model is so poor at predicting background ozone, does it make sense to evaluate the contribution of halogens to ozone depletion events in this region?

6. In the Antarctic, another source of bromine activation that has not been included here may be more important (from sea-salt aerosols formed form blowing snow, Yang et al., 2010). The authors should discuss more clearly the implications for not included this mechanism, which may be included in a future study.

Minor comments:

1. Abstract – "Most likely, they are related to events of boundary layer enhancement of bromine." This statement doesn't accurately reflect our understanding of boundary layer ozone depletion events, suggest to take out "Most likely".

2. Page 1 – Line 13: "Events of near-complete depletion of polar boundary layer ozone are observed frequently during spring-time over both hemispheres (Oltmans, 1981; Bottenheim et al., 1986, 2002, 2009)". I expect to see Barrie et al. as a main reference in this reference list.

3. Page 5 - Line 28: This sentence should be combined with next paragraph to avoid having a one sentence paragraph.

References:

Abbatt, J. P. D., Thomas, J. L., Abrahamsson, K., Boxe, C., Granfors, A., Jones, A. E., King, M. D., Saiz-Lopez, A., Shepson, P. B., Sodeau, J., Toohey, D. W., Toubin, C., von Glasow, R., Wren, S. N., and Yang, X.: Halogen activation via interactions with environmental ice and snow in the polar lower troposphere and other regions, Atmos. Chem. Phys., 12, 6237-6271, https://doi.org/10.5194/acp-12-6237-2012, 2012.

Barrie, L. A., Bottenheim, J. W., Schnell, R. C., Crutzen, P. J., and Rasmussen, R. A.: Ozone destruction and photochemical reactions at polar sunrise in the lower Arctic atmosphere, Nature, 334, 138–141, doi:10.1038/334138a0, 1988.

Pratt, K. A., Custard, K. D., Shepson, P. B., Douglas, T. A., Pöhler, D., General, S., Zielcke, J., Simpson, W. R., Platt, U., Tanner, D. J., Huey, L. G., Carlsen, M., and Stirm, B. H.: Photochemical production of molecular bromine in Arctic surface snowpacks, Nat. Geosci., 6, 351–356, doi:10.1038/NGEO1779, 2013.

Simpson, W. R., von Glasow, R., Riedel, K., Anderson, P., Ariya, P., Bottenheim, J., Burrows, J., Carpenter, L. J., Frieß, U., Goodsite, M. E., Heard, D., Hutterli, M., Jacobi, H.-W., Kaleschke, L., Neff, B., Plane, J., Platt, U., Richter, A., Roscoe, H., Sander, R., Shepson, P., Sodeau, J., Steffen, A., Wagner, T., and Wolff, E.: Halogens and their

role in polar boundary-layer ozone depletion, Atmos. Chem. Phys., 7, 4375-4418, https://doi.org/10.5194/acp-7-4375-2007, 2007.

Yang, X., Pyle, J. A., Cox, R. A., Theys, N., and Van Roozendael, M.: Snow-sourced bromine and its implications for polar tropospheric ozone, Atmos. Chem. Phys., 10, 7763–7773, doi:10.5194/acp-10-7763-2010, 2010.

———————————————

---

## Author Comment (AC1) · 11 Oct 2017

**Authors' response**

gmd-2017-126-RC1-supplement (2 August 2017)

We thank the anonymous referee #1 for the comments regarding our paper. We appreciate suggestions for further studies. Nevertheless, we would like to stress, that the main purpose of this paper is providing a proper description and reference regarding the implementation of the bromine explosion mechanism in EMAC. Although we compare modeling results with observational data, we do not study the mechanism, which had been proposed by Toyota et al. (2011), in detail.

- General comments:

  - Examination of TIMING of the bromine release mechanism for better understanding of the proposed/applied mechanism: Get corresponding model BrO VCD according to satellite overpass time for chosen sites. Scatter plot based on whole year data as in Yang et al. (2010). For better comparison, a lead-lag relationship can be used. We much appreciate the proposal of further statistical analysis regarding temporal coincidence of BrO enhancements comparing GOME satellite observed VCD and our modeling results. The modeling data will indeed allow for a variety of studies, e.g. temporal or spatial correlations as proposed by the referee. We would like to address these in follow-up studies. Here, as stated above, we intent to focus on a proper deception of the mechanism and its implementation into EMAC to serve as reference. A detailed validation of the mechanism in comparison to observation is beyond the scope of the present manuscript. Closely following the work of Toyota et al. (2011), we are able to show that the mechanism works astonishingly well without any change of parameter or fine-tuning to our model. We provide here a figure (Fig. 1) of BrO VCD at the sites which had been chosen for ODE. However, while this provides some comparison of timing of BrO enhancements, we acknowledge that this is not a proper validation and choose not to include this figure in our manuscript. To assess the contribution of bromine explosions to the BrO VCD, we subtracted the reference simulation (BrXplo_ref) from model integrations including bromine explosions (BrXplo_fysic, BrXplo_mysic, and BrXplo_mysic_rs). A computed zonal mean BrO VCD has been subtracted from GOME tropospheric VCD to highlight bromine explosion events. Satellite data, however does not allow for assessing the most interesting dates in northern and southern hemispheric winter where model results show a strong enhancement of BrO and ODE not present in surface ozone observation. We find a general but not strict temporal agreement in case of Barrow in spring-time. In late April and early May, we do not find BrO enhancements at Alert. Since our modeling results have not shown the long-lasting 2000s ODE at Alert this was to be expected. As pointed out by Strong et al. (2002), this long-lasting depletion event was related to transport of ozone poor air originating from sea ice. It is not clear whether transport or depletion is too weak in our simulation. Some better agreement is found in case of Zeppelin Mountain. At Neumeyer Station we probably find a coincidental event in late September. If ODE are qualitatively well reproduced in comparison with observation, we do also find coincidental BrO enhancements. But studying these in detail is well beyond the scope of this paper.

  - Missing sea spray acting as a bromine source. Why this kind of source is not included in the EMAC? A discussion covering this issue should be given. In general, an emission of tracers from sea spray could be included in EMAC. Since we focus on the implementation of a simple bromine release mechanism from sea ice, we have not considered sea spray on purpose. However, as pointed out by the referee, we shall include a discussion about this matter in the revised manuscript.

- Specific comments:

  - P2 L1: a review paper by Abbatt et al. (2012) should be cited here. We thank the referee for suggesting to include a reference to the work of Abbatt et al. (2012).

  - P2 L22: removal the pair of bracket in "(boundary)" By putting brackets in "(boundary)" we intended to acknowledge the capability of EMAC treating input data at *any* given level not only at the boundary layer as source of emission. We removed the brackets and made our point clearer: *[...] concentrations of tracers at the boundary layer or any other given level [...]*

[Figure]

**Figure 1.** BrO VCD evaluated at observation sites on the northern southern hemisphere. The same as for surface ozone comparison have been chosen. The reference simulation is subtracted from the shown modeling results. GOME tropospheric VCD is shown subtracted by its zonal mean to emphasize BrO enhancements.

- **P2 L23: why italic "online" is used here?** The comment is absolutely valid. For there is no specific reason, we removed the italic font.

- **P3 L20–21: Some discussions should be given to explain why such as a higher value (7.5%) of molar yield at solar zenith angle $> 85°$ (comparing to 0.1% at dark) is introduced in the model, though this number is from Toyota et al paper. This parameter is one critical parameter to allow enough bromine releasing from snow to match the observation. Either a justification, e.g. reference, or a caution must be given to remind readers of what is going on here.** It has to be indeed remarked that these values are of importance to the amount of $Br_2$ released by the mechanism. As pointed out, the values of $\Phi_1$ have been taken from Toyota et al. (2011) and have not been *tuned* to our model. In Section 3.1, Toyota et al. (2011) describe in detail how they obtain the specific value of 0.075 though cross-validation with observed spring-time ozone boundary layer values at Alert, Barrow, and Zeppelin. We add this reminder: *The specific value of $\Phi_1$ has been cautiously obtained as best choice by cross-validating modeling results with observed spring-time boundary layer ozone data at Alert, Barrow, and Zeppelin (Toyota et al., 2011, Section 3.1).*

- **P6 figure 2 and P7 L1–2: is the EMAC BrO VCD shown here a total of tropospheric and stratospheric BrO? If so, then a tropospheric column value should be worked out to make a direct comparison with satellite-based tropospheric BrO.** We have indeed compared *tropospheric* GOME BrO VCD with *total* BrO VCD of our model simulation. Total GOME BrO VCD have now been provided by courtesy of Andreas Richter (University of Bremen). We update all figures and comparisons in the corresponding section accordingly.

- **P7 L1–9: as mentioned in the general comment, just a spatial comparison for BrO is not good enough, a temporal comparison between daily satellite BrO VCD and corresponding model BrO should be given here to allow a further examination of the bromine releasing mechanism applied.** The purpose of our current paper is not to examine the release mechanism originally proposed by Toyota et al. (2011), but to implement the mechanism in our model. The suggested comparisons may be subject to further, more detailed studies.

**gmd-2017-126-RC2-interactively (4 August 2017)**

We would like to thank the anonymous referee #2 for suggesting further important literature and elaboration of sections.

– Major comments:

- **How is bromine recycling on aerosol treated? Is this important to sustain halogen activation and does it contribute to ozone depletion events?** Bromine recycling on aerosols is treated in the same way as it is for polar stratospheric clouds (PSCs). In these cold regimes, icy surfaces allow or accelerate reactions which are impossible or rather slow in the gas phase. For sustaining catalytic ozone depletion, the activation of halogens through heterogeneous reactions is very important. The set of heterogeneous reactions involving bromine and chlorine used in our simulation is given in the updated supplement.

- **The authors should look deeper into the literature as to how the understanding of halogen chemistry in the Arctic & Antarctic has developed over time. Papers such as Barrie et al. (1988) and Abbatt et al. (2012) should not be omitted from the reference list. In addition, Simpson et al. (2007) provides an excellent overview of how our understanding of halogen chemistry and ODEs has developed.** We much appreciate the suggestion of these important papers and take them into consideration in our revised introduction.

- **A clearer discussion of how snow contributes to halogen activation is needed, as discussed by Pratt et al. (2013) and Thomas et al., 2011.** Thank you for pointing this out. We will include a discussion of halogen activation on snow in our revised manuscript.

- **A list of the reactions that are included to describe the halogen cycle is needed either in the paper or in the supplement, including a short discussion of how heterogeneous reactions on aerosols are treated.** A list of heterogeneous reactions as implemented in the model have been added as supplement. A discussion will be included in the revised introduction.

- In general, I find the discussion of the results too short. Major features of the figures are not really described, which leaves the reader a bit lost as to what the model validation section means. For example, why is the surface ozone so low in the model compared to the measurement sites in Antarctica (Neumeyer and South Pole Stations)? If the model is so poor at predicting background ozone, does it make sense to evaluate the contribution of halogens to ozone depletion events in this region? Since the description and discussion of the figures and results may be indeed slightly too brief, we will add a more thorough description of the plots and their features. Regarding the prediction capabilities of surface ozone in Antarctica, although the model prediction is systematically below observation in the southern hemisphere and Greenland, it is appropriate to qualitatively look at the occurrence of ODEs there and how Antarctic ODEs are reproduced by this simple mechanism. There may be missing sources of ozone emission from the snowpack itself which are currently not implemented in EMAC. However, the intention of this manuscript is to describe the implemented bromine release mechanism, not a general validation of the model performance.

- In the Antarctic, another source of bromine activation that has not been included here may be more important (from sea-salt aerosols formed form blowing snow, (Yang et al., 2010)). The authors should discuss more clearly the implications for not included this mechanism, which may be included in a future study. We acknowledge the work by Yang et al. (2010) and the importance of the blowing-snow that has been neglected in our model so far. We will include this in the revised discussion. Bearing in mind that the release of sea salt aerosols is not included in our model simulations, we believe that it is nevertheless instructive to test by how much the Toyota et al. (2011) mechanism can explain bromine enhancements in Southern Hemisphere high latitudes.

– Minor comments:

- Abstract – "Most likely, they are related to events of boundary layer enhancement of bromine." This statement doesn't accurately reflect our understanding of boundary layer ozone depletion events, suggest to take out "Most likely". We follow the suggestion of the referee.

- P1 L13: "Events of near-complete depletion of polar boundary layer ozone are observed frequently during spring-time over both hemispheres (Oltmans, 1981; Bottenheim et al., 1986, 2002, 2009)". I expect to see Barrie et al. as a main reference in this reference list. We have included a citation of the important work by Barrie et al. (1988).

- P5 L28: This sentence should be combined with next paragraph to avoid having a one sentence paragraph. We follow the suggestion.

**References**

Abbatt, T. P. D., Thomas, J. L., Abrahamsson, K., Boxxe, C., Granfors, A., Jones, A. E., King, M. D., Saiz-Lopez, A., Shepson, P. B., Sodeau, J., Toohey, D. W.and Toubin, C., von Glasow, R., Wren, S. N., and Yang, X.: Halogene activation via interactions with environmental ice and snow in the polar lower troposphere amd other regions, Atmos. Chem. Phys., 12, 6237–6271, doi:10.5194/acp-12-6237-2012, 2012.

5  Barrie, L. A., Bottenheim, J. W., Schnell, R. C., Crutzen, P. J., and Rasmussen, R. A.: Ozone destruction and photochemical reactions at polar sunrise in the lower Arctic atmosphere, Nature, 334, 138–141, doi:10.1038/334138a0, 1988.

Bottenheim, J. W., Gallant, A. G., and Brice, K. A.: Measutements of NOy Species and O-3 at 82-Degrees-N Latitude, Geophys. Res. Lett., 13, 113–116, doi:10.1029/GL013i002p00113, 1986.

Bottenheim, J. W., Fuentes, J. D., Tarasick, D. W., and Anlauf, K. G.: Ozone in the Arctic lower troposphere during winter and spring 2000
10  (ALERT2000), Atmos. Environ., 36, 2535–2544, doi:10.1016/S1352-2310(02)00121-8, 2002.

Bottenheim, J. W., Netcheva, S., Morin, S., and Nghiem, S. V.: Ozone in the boundary layer air over the Arctic Ocean: measurements during the TARA transpolar drift 2006-2008, Atmos. Chem. Phys., 9, 4545–4557, 2009.

Oltmans, S. J.: Surface Ozone Measurements In Clean-Air, J. Geophys. Res.-Oceans Atmos., 86, 1174–1180, doi:10.1029/JC086iC02p01174, 1981.

15  Pratt, K. A., Custard, K. D., Shepson, P. B., Douglas, T. A., Pöhler, D., General, S., Zielcke, J., Simpson, W. R., Platt, U., Tanner, D. J., Huey, L. G., Carlsen, M., and Stirm, B. H.: Photochemical production of molecular bromine in Arctic surface snowpacks, Nat. Geosci., 6, 351–356, doi:10.1038/NGEO1779, 2013.

Simpson, W. R., von Glasow, R., Riedel, K., Anderson, P., Ariya, P., Bottenheim, J., Burrows, J., Carpenter, L. J., Friess, U., Goodsite, M. E., Heard, D., Hutterli, M., Jacobi, H.-W., Kaleschke, L., Neff, B., Plane, J., Platt, U., Richter, A., Roscoe, H., Sander, R., Shepson, P.,
20  Sodeau, J., Steffen, A., Wagner, T., and Wolff, E.: Halogens and their role in polar boundary-layer ozone depletion, Atmos. Chem. Phys., 7, 4375–4418, 2007.

Strong, C., Fuentes, J. D., Davis, R. E., and Bottenheim, J. W.: Thermodynamic attributes of Arctic boundary layer ozone depletion, Atmos. Environ., 36, 2641–2652, doi:https://doi.org/10.1016/S1352-2310(02)00114-0, http://www.sciencedirect.com/science/article/pii/S1352231002001140, Air/Snow/Ice Interactions in the Arctic: Results from ALERT 2000 and SUMMIT 2000, 2002.

25  Toyota, K., McConnell, J. C., Lupu, A., Neary, L., McLinden, C. A., Richter, A., Kwok, R., Semeniuk, K., Kaminski, J. W., Gong, S. L., Jarosz, J., Chipperfield, M. P., and Sioris, C. E.: Analysis of reactive bromine production and ozone depletion in the Arctic boundary layer using 3-D simulations with GEM-AQ: inference from synoptic-scale patterns, Atmos. Chem. Phys., 11, 3949–3979, doi:10.5194/acp-11-3949-2011, 2011.

Yang, X., Pyle, J. A., Cox, R. A., Theys, N., and Van Roozendael, M.: Snow-sourced bromine and its implications for polar tropospheric
30  ozone, Atmos. Chem. Phys., 10, 7763–7773, doi:10.5194/acp-10-7763-2010, 2010.

---

## Editor Comment (EC1) · S. Bekki (Editor) · 19 Oct 2017

I would like to remind the aims and scope of GMD to the authors The reviewers are asking for a more detailed and thorough evaluation of the model results against observations. Unfortunately, the authors tend to "kick in touch", stating that it is beyond the scope of the paper and that a thorough evaluation will be carried out in the following studies. For instance, the authors reply: "Nevertheless, we would like to stress, that the

main purpose of this paper is providing a proper description and reference regarding the implementation of the bromine explosion mechanism in EMAC". "Here, as stated above, we intent to focus on a proper description of the mechanism and its implementation into EMAC to serve as reference. A detailed validation of the mechanism in comparison to observation is beyond the scope of the present manuscript.". "The purpose of our current paper is not to examine the release mechanism originally proposed by Toyota et al. (2011), but to implement the mechanism in our model. The suggested comparisons may be subject to further, more detailed studies.". There might be a misunderstanding about GMD aims and scope. As stated on the 'about' page of the website, it is an "international scientific journal dedicated to the publication and public discussion of the description, development, AND evaluation of numerical models of the Earth system and its components". GMD articles are not just descriptions of numerical codes. The evaluation is an integral part of the papers. Just like the description, the evaluation should be as thorough as possible. I encourage the authors to follow the reviewers' recommendations regarding the evaluation. It would greatly strengthen the paper.

---

## Author Comment (AC2) · 8 Nov 2017

Dear Mr Bekki, we apologize in case we have been suggestive of being unwilling to meet the aims of GMD in our previous response. We have taken all referees' remarks seriously and are working on a satisfying revision. Due to the extensiveness of additional analysis and other commitments we may need more time to meet our high standards before resubmitting. Sincerely Yours, Stefanie Falk in behalf of all authors

---

## Author Response (AR1)

**Summary of changes in the revised manuscript**

Following the reviewers' suggestions, we are now comparing anomalies of vertical column densities of $BrO$ between GOME and our EMAC simulations in Fig. 4, rather than with the derived GOME-SLIMCAT tropospheric $BrO$ product. In addition to the anomaly plots highlighting the bromine explosion events, we provide comparisons of zonal mean absolute $BrO$ vertical columns densities as supplementary information.

We appreciate the reviewers' suggestions to look at a statistical correlation of observed and measured $BrO$ VCDs on a day-to-day basis, but unfortunately we don't have the daily data at hand to do a meaningful comparison here. Instead, we have extended the comparison between modelled and measured surface ozone including a statistical (lag-)correlation at selected stations.

We have also included a number of specific changes that we list in the following, section by section. The detailed text changes are color coded in the revised manuscript attached. Wording has been corrected where pointed out by the reviewers. Also some typographic mistakes have been corrected.

**Section 1**

– Following the suggestions of the reviewers additional citations, e.g., Barrie et al. (1988) and Abbatt et al. (2012), have been included.

– Based on the review article by Abbatt et al. (2012), we have extended our introduction regarding similarities between polar boundary layer heterogeneous chemistry and PSCs

– as well as a more detailed description of the prominent bromine release mechanism categories (frost flowers, bulk ice and snow, blowing snow, snowpack chemistry).

**Section 2**

– We have added a remark on similarities in heterogeneous chemistry between the polar boundary layer and PSCs in accordance with Section 1.

– We provide a list of bromine related heterogeneous reactions included in MECCA as Supplement S.1.

– We have adding more detail about the software structure and a remark on the used heterogeneous chemistry.

**Section 3**

In accordance to the reviewers' comments we have made major revisions to this section:

– We have extended the description and discussion of the results of both $BrO$ and $O_3$.

– Due to this extension, we have decided splitting the section into two subsection for comparison of total $BrO$ VCD and ODE, respectively.

– Previously, we compared modeled total VCD with tropospheric VCD of GOME. By courtesy, Andreas Richter (University of Bremen) provided us with monthly averaged total $BrO$ VCD of GOME. To avoid uncertainties due to SLIMCAT retrieved tropospheric columns in case of GOME date as well as additional uncertainties by estimating modeled tropospheric VCD, we have chosen a comparison of GOME and EMAC with respect to a total $BrO$ VCD. This also allows for an analysis as unbiased as possible for the datasets are most similar and can be further processed in the exact same way.

– Spatial comparison (Fig. 4) of observation and model is now displayed as anomalies with respect to the zonal mean emphasizing the $BrO$ hotspots.

– Accordingly, the description of the results has been extended.

- Additionally, for both, observation and model (BrXplo_ref, BrXplo_mysic), we have computed zonal means (Supplement S.5). We show that the overall model performance regarding VCD of $BrO$ is is improved by applying this simple bromine release mechanism.

- The chosen data, however, does not provide a temporal resolution better than 1-monthly. For addressing the reviewers' comments regarding a temporal correlation of events between observation and model, we have chosen our ozone data that is available in 1-hourly resolution.

- We have computed correlation coefficients at Barrow between surface ozone observation and two model integrations. In Fig. 7, the correlation is shown as binned 2D-histogram. We provide additional plots for the stations in the northern hemisphere as Supplement S.7. Text has been added accordingly.

- We have added a remark about the recognized ODE at Alert in 2000, which is absent in our modeling results.

- We have appended the discussion regarding further release mechanisms such as sea spray or blowing snow has been.

- Results of the additional sensitivity study with reduced ozone dry deposition has been moved to this section.

**Section 4**

- The section has been extended in accordance to the above changes.

**Authors' response**

**gmd-2017-126-RC1-supplement (2 August 2017)**

We thank the anonymous referee #1 for the comments regarding our paper. We appreciate suggestions for further studies. Nevertheless, we would like to stress, that the main purpose of this paper is providing a proper description and reference regarding the implementation of the bromine explosion mechanism in EMAC. Although we compare modeling results with observational data, we do not study the mechanism, which had been proposed by Toyota et al. (2011), in detail.

- General comments:

  • Examination of TIMING of the bromine release mechanism for better understanding of the proposed/applied mechanism: Get corresponding model $BrO$ VCD according to satellite overpass time for chosen sites. Scatter plot based on whole year data as in Yang et al. (2010). For better comparison, a lead-lag relationship can be used. We much appreciate the proposal of further statistical analysis regarding temporal coincidence of $BrO$ enhancements comparing GOME satellite observed VCD and our modeling results. The modeling data will indeed allow for a variety of studies, e.g. temporal or spatial correlations as proposed by the referee. We would like to address these in follow-up studies. Here, as stated above, we intent to focus on a proper deception of the mechanism and its implementation into EMAC to serve as reference. A detailed validation of the mechanism in comparison to observation is beyond the scope of the present manuscript. Closely following the work of Toyota et al. (2011), we are able to show that the mechanism works astonishingly well without any change of parameter or fine-tuning to our model. We provide here a figure (Fig. 1) of $BrO$ VCD at the sites which had been chosen for ODE. However, while this provides some comparison of timing of $BrO$ enhancements, we acknowledge that this is not a proper validation and choose not to include this figure in our manuscript. To assess the contribution of bromine explosions to the $BrO$ VCD, we subtracted the reference simulation (BrXplo_ref) from model integrations including bromine explosions (BrXplo_fysic, BrXplo_mysic, and BrXplo_mysic_rs). A computed zonal mean $BrO$ VCD has been subtracted from GOME tropospheric VCD to highlight bromine explosion events. Satellite data, however does not allow for assessing the most interesting dates in northern and southern hemispheric winter where model results show a strong enhancement of $BrO$ and ODE not present in surface ozone observation. We find a general but not strict temporal

agreement in case of Barrow in spring-time. In late April and early May, we do not find BrO enhancements at Alert. Since our modeling results have not shown the long-lasting 2000s ODE at Alert this was to be expected. As pointed out by Strong et al. (2002), this long-lasting depletion event was related to transport of ozone poor air originating from sea ice. It is not clear whether transport or depletion is too weak in our simulation. Some better agreement is found in case of Zeppelin Mountain. At Neumeyer Station we probably find a coincidental event in late September. If ODE are qualitatively well reproduced in comparison with observation, we do also find coincidental BrO enhancements. But studying these in detail is well beyond the scope of this paper.

- Missing sea spray acting as a bromine source. Why this kind of source is not included in the EMAC? A discussion covering this issue should be given. In general, an emission of tracers from sea spray could be included in EMAC. Since we focus on the implementation of a simple bromine release mechanism from sea ice, we have not considered sea spray on purpose. However, as pointed out by the referee, we shall include a discussion about this matter in the revised manuscript.

- Specific comments:

  - P2 L1: a review paper by Abbatt et al. (2012) should be cited here. We thank the referee for suggesting to include a reference to the work of Abbatt et al. (2012).

  - P2 L22: removal the pair of bracket in "(boundary)" By putting brackets in "(boundary)" we intended to acknowledge the capability of EMAC treating input data at *any* given level not only at the boundary layer as source of emission. We removed the brackets and made our point clearer: *[...] concentrations of tracers at the boundary layer or any other given level [...]*

  - P2 L23: why italic "online" is used here? The comment is absolutely valid. For there is no specific reason, we removed the italic font.

  - P3 L20–21: Some discussions should be given to explain why such as a higher value (7.5%) of molar yield at solar zenith angle $> 85°$ (comparing to 0.1% at dark) is introduced in the model, though this number is from Toyota et al paper. This parameter is one critical parameter to allow enough bromine releasing from snow to match the observation. Either a justification, e.g. reference, or a caution must be given to remind readers of what is going on here. It has to be indeed remarked that these values are of importance to the amount of $Br_2$ released by the mechanism. As pointed out, the values of $\Phi_1$ have been taken from Toyota et al. (2011) and have not been *tuned* to our model. In Section 3.1, Toyota et al. (2011) describe in detail how they obtain the specific value of 0.075 though cross-validation with observed spring-time ozone boundary layer values at Alert, Barrow, and Zeppelin. We add this reminder: *The specific value of $\Phi_1$ has been cautiously obtained as best choice by cross-validating modeling results with observed spring-time boundary layer ozone data at Alert, Barrow, and Zeppelin (Toyota et al., 2011, Section 3.1).*

  - P6 figure 2 and P7 L1–2: is the EMAC BrO VCD shown here a total of tropospheric and stratospheric BrO? If so, then a tropospheric column value should be worked out to make a direct comparison with satellite-based tropospheric BrO. We have indeed compared *tropospheric* GOME BrO VCD with *total* BrO VCD of our model simulation. Total GOME BrO VCD have now been provided by courtesy of Andreas Richter (University of Bremen). We update all figures and comparisons in the corresponding section accordingly.

  - P7 L1–9: as mentioned in the general comment, just a spatial comparison for BrO is not good enough, a temporal comparison between daily satellite BrO VCD and corresponding model BrO should be given here to allow a further examination of the bromine releasing mechanism applied. The purpose of our current paper is not to examine the release mechanism originally proposed by Toyota et al. (2011), but to implement the mechanism in our model. The suggested comparisons may be subject to further, more detailed studies.

**gmd-2017-126-RC2-interactively (4 August 2017)**

We would like to thank the anonymous referee #2 for suggesting further important literature and elaboration of sections.

[Figure]

**Figure 1.** BrO VCD evaluated at observation sites on the northern southern hemisphere. The same as for surface ozone comparison have been chosen. The reference simulation is subtracted from the shown modeling results. GOME tropospheric VCD is shown subtracted by its zonal mean to emphasize BrO enhancements.

- Major comments:

  - **How is bromine recycling on aerosol treated? Is this important to sustain halogen activation and does it contribute to ozone depletion events?** Bromine recycling on aerosols is treated in the same way as it is for polar stratospheric clouds (PSCs). In these cold regimes, icy surfaces allow or accelerate reactions which are impossible or rather slow in the gas phase. For sustaining catalytic ozone depletion, the activation of halogens through heterogeneous reactions is very important. The set of heterogeneous reactions involving bromine and chlorine used in our simulation is given in the updated supplement.

  - **The authors should look deeper into the literature as to how the understanding of halogen chemistry in the Arctic & Antarctic has developed over time. Papers such as Barrie et al. (1988) and Abbatt et al. (2012) should not be omitted from the reference list. In addition, Simpson et al. (2007) provides an excellent overview of how our understanding of halogen chemistry and ODEs has developed.** We much appreciate the suggestion of these important papers and take them into consideration in our revised introduction.

  - **A clearer discussion of how snow contributes to halogen activation is needed, as discussed by Pratt et al. (2013) and Thomas et al. (2011).** Thank you for pointing this out. We will include a discussion of halogen activation on snow in our revised manuscript.

  - **A list of the reactions that are included to describe the halogen cycle is needed either in the paper or in the supplement, including a short discussion of how heterogeneous reactions on aerosols are treated.** A list of heterogeneous reactions as implemented in the model have been added as supplement. A discussion will be included in the revised introduction.

  - **In general, I find the discussion of the results too short. Major features of the figures are not really described, which leaves the reader a bit lost as to what the model validation section means. For example, why is the surface ozone so low in the model compared to the measurement sites in Antarctica (Neumeyer and South Pole Stations)? If the model is so poor at predicting background ozone, does it make sense to evaluate the contribution of halogens to ozone depletion events in this region?** Since the description and discussion of the figures and results may be indeed slightly too brief, we will add a more thorough description of the plots and their features. Regarding the prediction capabilities of surface ozone in Antarctica, although the model prediction is systematically below observation in the southern hemisphere and Greenland, it is appropriate to qualitatively look at the occurrence of ODEs there and how Antarctic ODEs are reproduced by this simple mechanism. There may be missing sources of ozone emission from the snowpack itself which are currently not implemented in EMAC. However, the intention of this manuscript is to describe the implemented bromine release mechanism, not a general validation of the model performance.

  - **In the Antarctic, another source of bromine activation that has not been included here may be more important (from sea-salt aerosols formed form blowing snow, (Yang et al., 2010)). The authors should discuss more clearly the implications for not included this mechanism, which may be included in a future study.** We acknowledge the work by Yang et al. (2010) and the importance of the blowing-snow that has been neglected in our model so far. We will include this in the revised discussion. Bearing in mind that the release of sea salt aerosols is not included in our model simulations, we believe that it is nevertheless instructive to test by how much the Toyota et al. (2011) mechanism can explain bromine enhancements in Southern Hemisphere high latitudes.

- Minor comments:

  - **Abstract – "Most likely, they are related to events of boundary layer enhancement of bromine." This statement doesn't accurately reflect our understanding of boundary layer ozone depletion events, suggest to take out "Most likely".** We follow the suggestion of the referee.

  - **P1 L13: "Events of near-complete depletion of polar boundary layer ozone are observed frequently during springtime over both hemispheres (Oltmans, 1981; Bottenheim et al., 1986, 2002, 2009)". I expect to see Barrie et al. as a main reference in this reference list.** We have included a citation of the important work by Barrie et al. (1988).

- P5 L28: This sentence should be combined with next paragraph to avoid having a one sentence paragraph. We follow the suggestion.

[revised manuscript text omitted]

---

## Author Response (AR2)

**Authors' response**

In our revised manuscript we have taken into account the reviewers' comments and largely adhere to the suggested changes. In the course of this revision, We have revised large parts of Sect. 3 and Sect. 4 with respect to clarity in writing, e.g. we have elaborated on the description and discussion of our results. In particular, we:

5    – have added the suggested citations of further important publications on bromine explosion and ozone depletion events,

     – acknowledge the various observation techniques (in situ, ground-based, air-borne and satellite remote sensing) in a more balanced way,

     – once again include polar maps of total VCD in addition to the VCD anomalies,

     – add the more detailed supplementary figures (S.4-S.5) as Appendix A (Figs. A1-A4),

10    – display the time-lagged correlation coefficients for surface ozone at Alert and Barrow.

A detailed account of all changes in accordance to the referees' comments is given below and followed by the track changes.

**gmd-2017-126-RC1 (28 November 2017)**

We thank referee #1 for the critical comments that we try to address with our revisions.

- However, the only information about the BrO temporal comparison is a statement in section 4 where they declared
15    "Although a lag analysis shows highest temporal correlation at zero lag between observation and model data at Barrow, there are still notable differences to other observations." To be honest, this is a very vague statement, and scientifically unacceptable. What is the actual relation coefficient at Barrow and is it statistically significant? How about coefficients at other sites? Reader deserves to having this information. I suggest the authors to supply more details regarding the BrO temporal relations in a much clear way, e.g. by showing the coefficients and adding some discussions. We have
20    addressed the above remarks in our revised manuscript. In Fig. 8, we now show the time-lagged relationships of observed surface ozone and our different model experiments (BrXplo_ref, BrXplo_mysic, and BrXplo_rs) for Alert and Barrow, respectively. The simulated data have been shifted with respect to observation. Therefore a positive lag indicates a later occurrence of low surface ozone in the model experiment. In case of Barrow, the correlation is improved by switching on the bromine explosion mechanism (see also Fig. 7). The time-lagged correlation peaks at a lag of about $2\,\mathrm{h}$. Updating
25    the ozone surface resistance over ice ($r_{\mathrm{O_3}}^{\mathrm{ice-snow}} = 10000\,\mathrm{sm^{-1}}$) further increases the correlation. In case of Alert, the correlation peaks between $-40$ and $-48\,\mathrm{h}$ regardless, which means low ozone values occur about 2 days ahead of time in all model experiments. There is no improvement by switching on the bromine release mechanism. We conclude, the mechanism transmuting HBr, HOBr, and $\mathrm{BrNO_3}$ to $\mathrm{Br_2}$ sufficiently parameterized the main traits leading to the depletion of surface ozone in the Barrow region, but does not describe the situation at Alert well.

30 ### gmd-2017-126-RC2 (4 January 2018)

We would like to thank the anonymous referee #2 for advising further topic related publications. We appreciate the referees' remarks on the elaboration of sections and figures.

     – General comments:

- Note – line and figure numbering refers to the track changes version of the paper. There is a problem with the figure
35    numbering in this version as the first figure given is Figure 2. Indeed, we have mingled the consecutive numbering of the manuscript and the responses in the track changes version. However, the figure numbering in the final version of the manuscript is fine (for LATEX is handling it well).

- Conclusions – a comparison/discussion of results from other bromine activation mechanisms is needed. Specifically, a discussion of mechanism from X. Yang and its prior evaluation compared to GOME-2 is needed (e.g. Theys et al., ACP, 2011, DOI:10.5194/acp-11-1791-2011 and prior work). Where are the two different mechanisms working best, where are they failing? Is it necessary to use both mechanisms in a future study to evaluate them further? We have revised Section 4 with respect to the proposed questions.

- General – the authors should re-edit the paper for clarity as some of the writing is still quite unclear as to the meaning of the sentences. Thank you for pointing this out. We have thoroughly re-edit the manuscript locating sentences which might have been unclear.

– Specific comments:

- P1 L3: In the abstract & elsewhere in the paper. The authors seem to think the main evidence for boundary layer bromine in the polar regions is from satellites. This is not the case. The main evidence for boundary layer bromine is from in-situ and ground based remote sensing data. In addition, aircraft data in the troposphere provides direct evidence of the presence of bromine (e.g. during POLARCAT, ARCTAS). Then, satellites have been used the provide the spatial scale of such enhancements, but are relatively insensitive to boundary layer enhancements of bromine. The authors should clarify their discussion of measurements to more accurately reflect the knowledge of bromine enhancements and why they use satellite data to validate their model. I am not asking the authors to use other data in the present study, but to recognize that other data exists and should be used in a 2nd step to evaluate and further improve the model. We have added references to the POLARCAT and ARCTAS campaign. We have had no intention to neglect the comprehensive data collected through ground-based in situ as well as remote sensing observations of BrO in the Arctics. From the perspective of global modeling, global BrO VCD data from satellite remote sensing have been a natural choice for evaluation of modeling results. In addition, our starting point has been the long standing discrepancy between satellite observed and modeled tropospheric BrO, especially at high latitudes. Since there have been many studies focusing on particular observation sites, we intended to add a global perspective to the discussed mechanism (Toyota et al., 2011). Nevertheless, we have made use of ground-based observations' ozone data provided by various institutions in 1 hourly resolution (Section 3.2). We agree that in a second step also ground-based and air-borne BrO observations should be taken into consideration.

- P1 L19: Bromine explosion events "may" play an important role in mercury deposition. It is not 100% clear this is the case, so insert the word may here. Additional up to date references are needed for this sentence. We have chosen do cite recent modeling works by Toyota et al. (2014b, a) regarding the mercury deposition in the arctic and its connection to bromine explosions.

- P2 L3: Custard et al. is not a review paper. Thank you for pointing this out.

- P2 L17–25: This paragraph shows the authors have not thought about the difference between the stratosphere and knowledge of chemistry on ice surfaces at temperatures relevant to conditions in the lower troposphere. Specifically:

  - Ice above $-40°$ C has a liquid like layer, which does not behave as a gas-solid reactive surface. Thank you for pointing this out. We have revised the phrases that could have been misunderstood.

  - Multi-year ice has not been 100% discarded as a source of bromine and/or a surface on which bromine can be recycled once activated. We very much appreciate this remark, for it had been apparent from all referred papers that multi-year sea ice can by fully discarded as source. We hence re-write: *Therefore, multi-year sea ice contains much less salt than first year ice and may be discarded as a major source of BE events.*

  - It is not true that acidity is not important on ice surfaces. The study used as an example here is for HOBr uptake onto frozen NaBr/NaCl solutions. However, other studies for other surface reactions have shown a dependence on the pH of the solution before freezing (e.g. Oldridge and Abbatt, JCP A, 2011, DOI: 10.1021/jp200074u). In addition, this reference should be included at the end of line 25. We have added the mentioned reference (Oldridge and Abbatt, 2011) and re-wrote the sentence accordingly: *The importance of acidity for reaction kinetics on icy surfaces is strongly dependent on the involved species. While* HOBr *uptake on frozen* NaBr/NaCl

*solutions is not dependent on acidity (Adams et al., 2002), uptake reactions of gas-phase $O_3$ are fastest on acidic media (Oldridge and Abbatt, 2011).*

- The sentence on lines 23–25 has no clear meaning. We have re-written the sentence to make a clearer statement: *Ozone itself has the capacity of triggering auto-catalytic reactions by oxidizing bromine in snow and ice non-photochemically. Subsequently, $Br_2$ is released.*

- P2 L29–30: A more accurate and clear sentence is needed here. We have re-written: *Toyota et al. (2011) have shown that many aspects of observed bromine enhancements and boundary layer ODE can be reproduced with their simple approach of recycling $HOBr$, $BrNO_3$, and $HBr$ into $Br_2$.*

- P3 L5: Annual cycle "in" both hemispheres. Fix on -> in. We have corrected this typo on this and some other lines.

- P3 L21: Should cite Toyota's paper here, since this is where the mechanism came from. We have included the citation as follows: *Toyota et al. (2011) assume that [...]*

- P5: Note that in the track changes version, the first figure is Figure 2. This should be fixed in the final version. As mentioned above, the figure numbering is fine in the final version.

- P5 L21: Standard PSC chemistry is likely not sufficient for aerosols in the troposphere. This does not need to be fixed for this study, but it should be noted and mentioned as an area to work on later during the future work/conclusions/perspectives. We have added: *[...] we make use of EMAC's standard atmospheric bromine chemistry (Supplement S.1) that has been optimized for stratospheric conditions [...]. This treatment, however, might not be fully sufficient with respect to tropospheric heterogeneous chemistry on aerosols and should by subject to future work.* Furthermore, a discussion has been included in the revision of Section 4.

- P6 L26–27: Please spell out units of the deposition velocity so there is no confusion. There was indeed a confusion. Despite we wrote about the surface resistance, we have stated deposition velocities but with the wrong units. We have now corrected this faux pas.

- P6 L32: This is one of the areas where my first comment is important, see above. We have rephrased this passage and clarified our intentions for using $BrO$ VCD satellite data: *At first, we assess the spatial distribution of $BrO$ total VCD on global scales. Therefore, we compare EMAC (BrXplo_mysic) with GOME retrieved total VCD in both hemispheres (i.a., Wagner and Platt, 1998; Richter et al., 1998). Afterwards, we showcase implications on ODE regarding their temporal occurrence at specific ground-based observation sites in both hemispheres (Table 2).*

- Section 3.1: the writing/figures in this section need some work. Here are some specific suggestions:

  - Move Figure 1 from the response to reviewers into the paper, this is additional info that can be discussed in relationship to the ozone time series presented. Although Fig. 1 does provide additional information, we feel that there is a danger of misinterpretation of this analysis. Consequently, we choose to include this Figure neither in the Supplement nor in the main paper. Instead, we refer to a corresponding analysis in the paper of Toyota et al. (2011, e.g., Fig.10 and Fig.12).

  - I find the authors are discussing the figures in the supplement more than the figures in the paper, the authors should consider moving the more detailed supplement figures with BrO VCD from EMAC and GOME into the paper so the discussion is clearer. Both the VCD maps and the anomalies (shown now) should be in the paper. We have once again included the $BrO$ total VCD maps (as shown in a previous version of this manuscript) as Fig. 3. In the course of this, we revised the text with respect to clarity in writing and discussion of the Figs. 3-4. We have taken the suggestion of moving the supplementary Figures (S.4-S.5) to the paper into consideration. These supplementary figures are indeed more detailed regarding the full annual range of satellite observation and modeling results. We have thus included these figures as Appendix A (Figs. A1-A4).

  - Figure 4 (anomalies of VCD for BrO) - the months should be labeled on both the right and left panels. We have added labeling of the months on the right panels accordingly.

[revised manuscript text omitted]